# Private Hypothesis Selection

**Mark Bun**
Department of Computer Science
Boston University
mbun@bu.edu

**Gautam Kamath**
Cheriton School of Computer Science
University of Waterloo
g@csail.mit.edu

**Thomas Steinke**
IBM Research
phs@thomas-steinke.net

**Zhiwei Steven Wu**
Department of Computer Science & Engineering
University of Minnesota
zsw@umn.edu

## Abstract

We provide a differentially private algorithm for hypothesis selection. Given samples from an unknown probability distribution $P$ and a set of $m$ probability distributions $\mathcal{H}$, the goal is to output, in a $\varepsilon$-differentially private manner, a distribution from $\mathcal{H}$ whose total variation distance to $P$ is comparable to that of the best such distribution (which we denote by $\alpha$). The sample complexity of our basic algorithm is $O\left(\frac{\log m}{\alpha^2} + \frac{\log m}{\alpha\varepsilon}\right)$, representing a minimal cost for privacy when compared to the non-private algorithm. We also can handle infinite hypothesis classes $\mathcal{H}$ by relaxing to $(\varepsilon, \delta)$-differential privacy.

We apply our hypothesis selection algorithm to give learning algorithms for a number of natural distribution classes, including Gaussians, product distributions, sums of independent random variables, piecewise polynomials, and mixture classes. Our hypothesis selection procedure allows us to generically convert a cover for a class to a learning algorithm, complementing known learning lower bounds which are in terms of the size of the packing number of the class. As the covering and packing numbers are often closely related, for constant $\alpha$, our algorithms achieve the optimal sample complexity for many classes of interest. Finally, we describe an application to private distribution-free PAC learning.

## 1 Introduction

We consider the problem of *hypothesis selection*: given samples from an unknown probability distribution, select a distribution from some fixed set of candidates which is "close" to the unknown distribution in some appropriate distance measure. Such situations can arise naturally in a number of settings. For instance, we may have a number of different methods which work under various circumstances, which are not known in advance. One option is to run all the methods to generate a set of hypotheses, and pick the best from this set afterwards. Relatedly, an algorithm may branch its behavior based on a number of "guesses," which will similarly result in a set of candidates, corresponding to the output at the end of each branch. Finally, if we know that the underlying distribution belongs to some (parametric) class, it is possible to essentially enumerate the class (also known as a *cover*) to create a collection of hypotheses. Observe that this last example is quite general, and this approach can give generic learning algorithms for many settings of interest.

---

A full version of the paper, with additional details and proofs, appears in the supplement, or on arXiv [9].

This problem of hypothesis selection has been extensively studied (see, e.g., [46, 17, 18, 19]), resulting in algorithms with a sample complexity which is *logarithmic* in the number of hypotheses. Such a mild dependence is critical, as it facilitates sample-efficient algorithms even when the number of candidates may be large. These initial works have triggered a great deal of study into hypothesis selection with additional considerations, including computational efficiency, understanding the optimal approximation factor, adversarial robustness, and weakening access to the hypotheses (e.g., [36, 15, 16, 43, 2, 21, 1, 7]).

However, in modern settings of data analysis, data may contain sensitive information about individuals. Some examples of such data include medical records, GPS location data, or private message transcripts. As such, we would like to perform statistical inference in these settings without revealing significant information about any particular individual's data. To this end, there have been many proposed notions of data privacy, but perhaps the gold standard is that of *differential privacy* [26]. Informally, differential privacy requires that, if a single datapoint in the dataset is changed, then the distribution over outputs produced by the algorithm should be similar (see Definition 4). Differential privacy has seen widespread adoption, including deployment by Apple [22], Google [28], and the US Census Bureau [14].

This naturally raises the question of whether one can perform hypothesis selection under the constraint of differential privacy, while maintaining a logarithmic dependence on the size of the cover. Such a tool would allow us to generically obtain private learning results for a wide variety of settings.

## 1.1 Results

Our main results answer this in the affirmative: we provide differentially private algorithms for selecting a good hypothesis from a set of distributions. The output distribution is competitive with the best distribution, and the sample complexity is bounded by the logarithm of the size of the set. The following is a basic version of our main result.

**Theorem 1.** *Let $\mathcal{H} = \{H_1, \ldots, H_m\}$ be a set of probability distributions. Let $D = \{X_1, \ldots, X_n\}$ be a set of samples drawn independently from an unknown probability distribution $P$. There exists an $\varepsilon$-differentially private algorithm (with respect to the dataset $D$) which has following guarantees. Suppose there exists a distribution $H^* \in \mathcal{H}$ such that $d_{\mathrm{TV}}(P, H^*) \leq \alpha$. If $n = \Omega\left(\frac{\log m}{\alpha^2} + \frac{\log m}{\alpha \varepsilon}\right)$, then the algorithm will output a distribution $\hat{H} \in \mathcal{H}$ such that $d_{\mathrm{TV}}(P, \hat{H}) \leq (3+\zeta)\alpha$ with probability at least $9/10$, for any constant $\zeta > 0$. The running time of the algorithm is $O(nm^2)$.*

The sample complexity of this problem without privacy constraints is $O\left(\frac{\log m}{\alpha^2}\right)$, and thus the additional cost for $\varepsilon$-differential privacy is an additive $O\left(\frac{\log m}{\alpha \varepsilon}\right)$. We consider this cost to be minimal; in particular, the dependence on $m$ is unchanged. Note that the running time of our algorithm is $O(nm^2)$ – we conjecture it may be possible to reduce this to $\tilde{O}(nm)$ as has been done in the non-private setting [16, 43, 2, 1], though we have not attempted to perform this optimization. Regardless, our main focus is on the sample complexity rather than the running time, since any method for generic hypothesis selection requires $\Omega(m)$ time, thus precluding efficient algorithms when $m$ is large. Note that the approximation factor of $(3 + \zeta)\alpha$ is effectively tight [19, 36, 7]. Theorem 1 requires prior knowledge of the value of $\alpha$, though we can use this to obtain an algorithm with similar guarantees which does not (Theorem 3).

It is possible to improve the guarantees of this algorithm in two ways (Theorem 4). First, if the distributions are nicely structured, the former term in the sample complexity can be reduced from $O(\log m/\alpha^2)$ to $O(d/\alpha^2)$, where $d$ is a VC-dimension-based measure of the complexity of the collection of distributions. Second, if there are few hypotheses which are close to the true distribution, then we can pay only logarithmically in this number, as opposed to the total number of hypotheses. These modifications allow us to handle instances where $m$ may be very large (or even infinite), albeit at the cost of weakening to approximate differential privacy to perform the second refinement. A technical discussion of our methods is in Section 1.2, our basic approach is covered in Section 3, and the version with all the bells and whistles appears in Section 4.

From Theorem 1, we immediately obtain Corollary 1 which applies when $\mathcal{H}$ itself may not be finite, but admits a finite cover with respect to total variation distance.

**Corollary 1.** *Suppose there exists an $\alpha$-cover $\mathcal{C}_\alpha$ of a set of distributions $\mathcal{H}$, and that we are given a set of samples $X_1, \ldots, X_n \sim P$, where $d_{\mathrm{TV}}(P, \mathcal{H}) \leq \alpha$. For any constant $\zeta > 0$, there exists an $\varepsilon$-differentially private algorithm (with respect to the input $\{X_1, \ldots, X_n\}$) which outputs a distribution $H^* \in \mathcal{C}_\alpha$ such that $d_{\mathrm{TV}}(P, H^*) \leq (6 + 2\zeta)\alpha$ with probability $\geq 9/10$, as long as $n = \Omega\left(\frac{\log |\mathcal{C}_\alpha|}{\alpha^2} + \frac{\log |\mathcal{C}_\alpha|}{\alpha \varepsilon}\right)$.*

Informally, this says that if a hypothesis class has an $\alpha$-cover $\mathcal{C}_\alpha$, then there is a private learning algorithm for the class which requires $O(\log |\mathcal{C}_\alpha|)$ samples. Note that our algorithm works even if the unknown distribution is only *close* to the hypothesis class. This is useful when we may have model misspecification, or when we require adversarial robustness. The requirements for this theorem to apply are minimal, and thus it generically provides learning algorithms for a wide variety of hypothesis classes. That said, in non-private settings, the sample complexity given by this method is rather lossy: as an extreme example, there is no finite-size cover of univariate Gaussian distributions with unbounded parameters, so this approach does not give a finite-sample algorithm. That said, it is well-known that $O(1/\alpha^2)$ samples suffice to estimate a Gaussian in total variation distance. In the private setting, our theorem incurs a cost which is somewhat necessary: in particular, it is folklore that any pure $\varepsilon$-differentially private learning algorithm must pay a cost which is logarithmic in the packing number of the class. Due to the relationship between packing and covering numbers, this implies that up to a constant factor relaxation in the learning accuracy, our results are tight. A more formal discussion appears in the supplement.

Given Corollary 1, in Section 5, we derive new learning results for a number of classes. Our main applications are for $d$-dimensional Gaussian and product distributions. Informally, we obtain $\tilde{O}(d)$ sample algorithms for learning a product distribution and a Gaussian with known covariance, and an $\tilde{O}(d^2)$ algorithm for learning a Gaussian with unknown covariance (Corollaries 2 and 3). These improve on recent results by Kamath, Li, Singhal, and Ullman [33] in two different ways. First, as mentioned before, our results are semi-agnostic, so we can handle when the distribution is only *close* to a product or Gaussian distribution. Second, our results hold for pure $(\varepsilon, 0)$-differential privacy, which is a stronger notion than $\varepsilon^2$-zCDP as considered in [33]. In this weaker model, they also obtained $\tilde{O}(d)$ and $\tilde{O}(d^2)$ sample algorithms, but the natural modifications to achieve $\varepsilon$-DP incur extra $\mathrm{poly}(d)$ factors.[1] [33] also showed $\tilde{\Omega}(d)$ lower bounds for Gaussian and product distribution estimation in the even weaker model of $(\varepsilon, \delta)$-differential privacy. Thus, our results show that the dimension dependence for these problems is unchanged for essentially any notion of differential privacy. In particular, our results show a previously-unknown separation between mean estimation of product distributions and non-product distributions under pure $(\varepsilon, 0)$-differential privacy; see Remark 1.

We also apply Theorem 4 to obtain algorithms for learning Gaussians under $(\varepsilon, \delta)$-differential privacy, with no bounds on the mean and variance parameters. More specifically, we provide algorithms for learning multivariate Gaussians with unknown mean and known covariance (Corollary 4), in which we manage to avoid dependences which arise due to the application of advanced composition (similar to Remark 1). We additionally give algorithms for learning mixtures of any coverable class (Corollary 5). In particular, this immediately implies algorithms for learning mixtures of Gaussians, product distributions, and all other classes mentioned above. Additional classes we can learn privately, described and discussed in the supplement, include piecewise polynomials, sums of independent random variables, and univariate Gaussians with unbounded mean and variance.

To conclude our applications, we discuss a connection to PAC learning. It is known that the sample complexity of differentially private distribution-free PAC learning can be higher than that of non-private learning. However, this gap does not exist for distribution-specific learning, where the learning algorithm knows the distribution of (unlabeled) examples, as both sample complexities are characterized by VC dimension. Private hypothesis selection allows us to address an intermediate situation where the distribution of unlabeled examples is not known exactly, but is known to come (approximately) from a class of distributions. When this class has a small cover, we are able to recover sample complexity guarantees for private PAC learning which are comparable to the non-private case. Details and discussion appear in the supplement.

## 1.2 Techniques

Non-privately, most algorithms for hypothesis selection involve a tournament-style approach. We conduct a number of pairwise comparisons between distributions, which may either have a winner and a loser, or may be declared a draw. Intuitively, a distribution will be declared the winner of a comparison if it is much closer than the alternative to the unknown distribution, and a tie will be declared if the two distributions are comparably close. The algorithm will output any distribution which never loses a comparison. A single comparison between a pair of hypotheses requires $O(1/\alpha^2)$ samples, and a Chernoff plus union bound argument over the $O(m^2)$ possible comparisons increases the sample complexity to $O(\log m/\alpha^2)$. In fact, we can use uniform convergence arguments to reduce this sample complexity to $O(d/\alpha^2)$, where $d$ is the VC dimension of the $2\binom{m}{2}$ sets (the "Scheffé" sets) defined by the subsets of the domain where the PDF of one distribution dominates another. Crucially, we must reuse the same set of samples for all comparisons to avoid paying polynomially in the number of hypotheses.

A private algorithm for this problem requires additional care. Since a single comparison is based on the number of samples which fall into a particular subset of the domain, the sensitivity of the underlying statistic is low, and thus privacy may seem easily achievable at first glance. However, the challenge comes from the fact that the same samples are reused for all pairwise comparisons, thus greatly increasing the sensitivity: changing a single datapoint could flip the result of every comparison! In order to avoid this pitfall, we instead carefully construct a score function for each hypothesis, namely, the minimum number of points that must be changed to cause the distribution to lose any comparison. For this to be a useful score function, we must show that the best hypothesis will win all of its comparisons by a large margin. We can then use the Exponential Mechanism [37] to select a distribution with high score.

Further improvements can be made if we are guaranteed that the number of "good" hypotheses (i.e., those that have total variation distance from the true distribution bounded by $(3 + \zeta)\alpha$) is at most some parameter $k$, and if we are willing to relax to approximate differential privacy. The parameter $k$ here is related to the doubling dimension of the hypothesis class with respect to total variation distance. If we randomly assign the hypotheses to $\Omega(k^2)$ buckets, with high probability, no bucket will contain more than one good hypothesis. We can identify a bucket containing a good hypothesis using a similar method based on the exponential mechanism as described above. Moreover, since we are likely to only have one "good" hypothesis in the chosen bucket, this implies a significant gap between the best and second-best scores in that bucket. This allows us to use stability-based techniques [25, 44], and in particular the GAP-MAX algorithm of Bun, Dwork, Rothblum, and Steinke [8], to identify an accurate distribution.

## 1.3 Related Work

Our main result builds on a long line of work on non-private hypothesis selection. One starting point for the particular style of approach we consider here is [46], which was expanded on in [17, 18, 19]. Since then, there has been study into hypothesis selection under additional considerations, including computational efficiency, understanding the optimal approximation factor, adversarial robustness, and weakening access to the hypotheses [36, 15, 16, 43, 2, 21, 1, 7]. Our private algorithm examines the same type of problem, with the additional constraint of differential privacy.

There has recently been a great deal of interest in differentially private distribution learning. In the central model, most relevant are [20], which gives algorithms for learning structured univariate distributions, and [35, 33], which focus on learning Gaussians and binary product distributions. [13] also studies private statistical parameter estimation. Privately learning mixtures of Gaussians was considered in [39, 34]. The latter paper (which is concurrent with the present work) gives a computationally efficient algorithm for the problem, but with a worse sample complexity, and incomparable accuracy guarantees (they require a separation condition, and perform clustering and parameter estimation, while we do proper learning). [10] give an algorithm for learning distributions in Kolmogorov distance. Upper and lower bounds for learning the mean of a product distribution over the hypercube in $\ell_\infty$-distance include [6, 12, 26, 42]. [3] focuses on estimating properties of a distribution, rather than the distribution itself. [40] gives an algorithm which allows one to estimate asymptotically normal statistics with optimal convergence rates, but no finite sample complexity guarantees. There has also been a great deal of work on distribution learning in the local model

of differential privacy [23, 45, 32, 4, 24, 31, 47, 29]. Additional discussion of non-private learning appears in the supplementary material.

## 2 Preliminaries

**Definition 1.** *The* total variation distance *or* statistical distance *between $P$ and $Q$ is defined as $d_{\mathrm{TV}}(P, Q) = \max_{S \subseteq \Omega} P(S) - Q(S) = \frac{1}{2} \int_{x \in \Omega} |P(x) - Q(x)| dx = \frac{1}{2} \|P - Q\|_1 \in [0, 1]$. Moreover, if $\mathcal{H}$ is a set of distributions over a common domain, we define $d_{\mathrm{TV}}(P, \mathcal{H}) = \inf_{H \in \mathcal{H}} d_{\mathrm{TV}}(P, H)$.*

**Definition 2.** *A $\gamma$-cover of a set of distributions $\mathcal{H}$ is a set of distributions $\mathcal{C}_\gamma$, such that for every $H \in \mathcal{H}$, there exists some $P \in \mathcal{C}_\gamma$ such that $d_{\mathrm{TV}}(P, H) \leq \gamma$. A $\gamma$-packing of a set of distributions $\mathcal{H}$ is a set of distributions $\mathcal{P}_\gamma \subseteq \mathcal{H}$, such that for every pair of distributions $P, Q \in \mathcal{P}_\gamma$, we have that $d_{\mathrm{TV}}(P, Q) \geq \gamma$.*

In this paper, we present semi-agnostic learning algorithms.

**Definition 3.** *An algorithm is said to be an $\alpha$-semi-agnostic learner for a class $\mathcal{H}$ if it has the following guarantees. Suppose we are given $X_1, \ldots, X_n \sim P$, where $d_{\mathrm{TV}}(P, \mathcal{H}) \leq \mathrm{OPT}$. The algorithm must output some distribution $\hat{H}$ such that $d_{\mathrm{TV}}(P, H) \leq c \cdot \mathrm{OPT} + O(\alpha)$, for some constant $c \geq 1$. If $c = 1$, then the algorithm is said to be agnostic.*

We consider algorithms under the constraint of differential privacy.

**Definition 4** ([26]). *A randomized algorithm $T : X^* \to \mathcal{R}$ is $(\varepsilon, \delta)$-differentially private if for all $n \geq 1$, for all neighboring datasets $D, D' \in X^n$, and for all events $S \subseteq \mathcal{R}$, $\Pr[T(D) \in S] \leq e^\varepsilon \Pr[T(D') \in S] + \delta$. If $\delta = 0$, we say that $T$ is $\varepsilon$-differentially private.*

We will also use the related notion of concentrated differential privacy [27, 11], which is defined in the supplement.

Our methods will rely heavily upon the Exponential mechanism.

**Theorem 2** (Exponential Mechanism [37]). *For a score function $q : X^* \times \mathcal{R} \to \mathbb{R}$, define its sensitivity as $\Delta(q) = \max_{r \in \mathcal{R}, D \sim D'} |q(D, r) - q(D', r)|$. For any $q$, input data set $D$, and privacy parameter $\varepsilon > 0$, the exponential mechanism $\mathcal{M}_E(D, q, \varepsilon)$ picks an outcome $r \in \mathcal{R}$ with probability proportional to $\exp(\varepsilon q(D, r)/(2\Delta(q)))$. This is $\varepsilon$-differentially private, and with probability at least $1 - \beta$, selects an outcome $r \in \mathcal{R}$ such that $q(D, r) \geq \max_{r' \in \mathcal{R}} q(D, r') - \frac{2\Delta(q) \log(|\mathcal{R}|/\beta)}{\varepsilon}$.*

## 3 A First Method for Private Hypothesis Selection

In this section, we present our first algorithm for private hypothesis selection and obtain Theorem 1.

Note that the sample complexity bound above scales logarithmically with the size of the hypothesis class. In Section 4, we will provide a stronger result (which subsumes the present one as a special case) that can handle certain infinite hypothesis classes. For sake of exposition, we begin in this section with the basic algorithm.

### 3.1 Pairwise Comparisons

We first present a subroutine which compares two hypothesis distributions. Let $H$ and $H'$ be two distributions over domain $\mathcal{X}$ and consider the following set, which is called the *Scheffé set* $\mathcal{W}_1 = \{x \in \mathcal{X} \mid H(x) > H'(x)\}$. Define $p_1 = H(\mathcal{W}_1)$, $p_2 = H'(\mathcal{W}_1)$, and $\tau = P(\mathcal{W}_1)$ to be

the probability masses that $H$, $H'$, and $P$ place on $\mathcal{W}_1$, respectively. It follows that $p_1 > p_2$ and $p_1 - p_2 = d_{\text{TV}}(H, H')$.[2]

---

**Algorithm 1:** PAIRWISE CONTEST: PC$(H, H', D, \zeta, \alpha)$

**Input**: Two hypotheses $H$ and $H'$, input dataset $D$ of size $n$ drawn i.i.d. from target distribution $P$, approximation parameter $\zeta > 0$, and accuracy parameter $\alpha \in (0, 1)$.
**Initialize**: Compute the fraction of points that fall into $\mathcal{W}_1$: $\hat{\tau} = \frac{1}{n} |\{x \in D \mid x \in \mathcal{W}_1\}|$.
**If** $p_1 - p_2 \leq (2 + \zeta)\alpha$, return "Draw".
**Else If** $\hat{\tau} > p_1 - (1 + \zeta/2)\alpha$, return $H$ as the winner.
**Else If** $\hat{\tau} < p_2 + (1 + \zeta/2)\alpha$, return $H'$ as the winner.
**Else** return "Draw".

---

Now consider the following function of this ordered pair of hypotheses:

$$\Gamma_\zeta(H, H', D) = \begin{cases} n & \text{if } p_1 - p_2 \leq (2 + \zeta)\alpha; \\ n \cdot \max\{0, \hat{\tau} - (p_2 + (1 + \zeta/2)\alpha)\} & \text{otherwise.} \end{cases}$$

When the two hypotheses are sufficiently far apart (i.e., $d_{\text{TV}}(H, H') > (2 + \zeta)\alpha$), $\Gamma_\zeta(H, H', D)$ is essentially the number of points one needs to change in $D$ to make $H'$ the winner.

**Lemma 1.** *Let $P, H, H'$ be distributions as above. With probability at least $1 - 2\exp(-n\zeta^2\alpha^2/8)$ over the random draws of $D$ from $P^n$, $\hat{\tau}$ satisfies $|\hat{\tau} - \tau| < \zeta\alpha/4$, and if $d_{\text{TV}}(P, H) \leq \alpha$, then $\Gamma_\zeta(H, H', D) > \zeta\alpha n/4$.*

*Proof.* By applying Hoeffding's inequality, we know that with probability at least $1 - 2\exp(-n\zeta^2\alpha^2/8)$, $|\tau - \hat{\tau}| < \zeta\alpha/4$. We condition on this event for the remainder of the proof. Consider the following two cases. In the first case, suppose that $p_1 - p_2 \leq (2 + \zeta)\alpha$. Then we know that $\Gamma_\zeta(H, H', D) = n > \alpha n$. In the second case, suppose that $p_1 - p_2 > (2 + \zeta)\alpha$. Since $d_{\text{TV}}(P, H) \leq \alpha$, we know that $|p_1 - \tau| \leq \alpha$, and so $|p_1 - \hat{\tau}| < (1 + \zeta/4)\alpha$. Since $p_1 > p_2 + (2 + \zeta)\alpha$, we also have $\hat{\tau} > p_2 + (1 + 3\zeta/4)\alpha$. It follows that $\Gamma_\zeta(H, H', D) = n(\hat{\tau} - (p_2 + (1 + \zeta/2)\alpha)) > \zeta\alpha n/4$. This completes the proof. $\square$

## 3.2 Selection via Exponential Mechanism

In light of the definition of the pairwise comparison defined above, we consider the following score function $S\colon \mathcal{H} \times \mathcal{X}^n$, such that for any $H_j \in \mathcal{H}$ and dataset $D$, $S(H_j, D) = \min_{H_k \in \mathcal{H}} \Gamma_\zeta(H_j, H_k, D)$. Roughly speaking, $S(H_j, D)$ is the minimum number of points required to change in $D$ in order for $H_j$ to lose at least one pairwise contest against a different hypothesis. When the hypothesis $H_j$ is very close to every other distribution, such that all pairwise contests return "Draw," then the score will be $n$.

---

**Algorithm 2:** PRIVATE HYPOTHESIS SELECTION: PHS$(\mathcal{H}, D, \varepsilon)$

**Input**: Dataset $D$, a collection of hypotheses $\mathcal{H} = \{H_1, \ldots, H_m\}$, privacy parameter $\varepsilon$.
Output a random hypothesis $\hat{H} \in \mathcal{H}$ such that for each $H_j$, $\Pr[\hat{H} = H_j] \propto \exp\left(\frac{S(H_j, D)}{2\varepsilon}\right)$.

---

**Lemma 2** (Privacy). *For any $\varepsilon > 0$ and collection of hypotheses $\mathcal{H}$, the algorithm PHS$(\mathcal{H}, \cdot, \varepsilon)$ satisfies $\varepsilon$-differential privacy.*

*Proof.* First, observe that for any pairs of hypotheses $H_j, H_k$, $\Gamma_\zeta(H_j, H_k, \cdot)$ has sensitivity 1. As a result, the score function $S$ is also 1-sensitive. Then the result directly follows from the privacy guarantee of the exponential mechanism (Theorem 2). $\square$

**Lemma 3** (Utility). *Fix any $\alpha, \beta \in (0, 1)$, and constant $\zeta > 0$. Suppose that there exists $H^* \in \mathcal{H}$ such that $d_{\text{TV}}(P, H^*) \leq \alpha$. Then with probability $1 - \beta$ over the sample $D$ and the algorithm PHS,*

we have that $PHS(\mathcal{H}, D)$ outputs an hypothesis $\hat{H}$ such that $d_{\mathrm{TV}}(P, \hat{H}) \leq (3 + \zeta)\alpha$, as long as the sample size satisfies $n \geq \frac{8\ln(4m/\beta)}{\zeta^2\alpha^2} + \frac{8\ln(2m/\beta)}{\zeta\alpha\varepsilon}$.

*Proof.* First, consider the $m$ pairwise contests between $H^*$ and every candidate in $\mathcal{H}$. Let $\mathcal{W}_j = \{x \in \mathcal{X} \mid H_j(x) > H^*(x)\}$ be the collection of Scheffé sets. For any event $W \subseteq \mathcal{X}$, let $\hat{P}(W)$ denote the empirical probability of event $W$ on the dataset $D$. By Lemma 1 and an application of the union bound, we know that with probability at least $1 - 2m\exp(-n\zeta^2\alpha^2/8)$ over the draws of $D$, $|P(\mathcal{W}_j) - \hat{P}(\mathcal{W}_j)| \leq \zeta\alpha/4$ and $\Gamma_\zeta(H^*, H_j, D) > \zeta\alpha n/4$ for all $H_j \in \mathcal{H}$. In particular, the latter event implies that $S(H^*, D) > \zeta\alpha n/4$.

Next, by the utility guarantee of the exponential mechanism (Theorem 2), we know that with probability at least $1 - \beta/2$, the output hypothesis satisfies $S(\hat{H}, D) \geq S(H^*, D) - \frac{2\ln(2m/\beta)}{\varepsilon} > \zeta\alpha n/4 - \frac{2\ln(2m/\beta)}{\varepsilon}$. Then as long as $n \geq \frac{8\ln(4m/\beta)}{\zeta^2\alpha^2} + \frac{8\ln(2m/\beta)}{\zeta\alpha\varepsilon}$, we know that with probability at least $1 - \beta$, $S(\hat{H}, D) > 0$. Let us condition on this event, which implies that $\Gamma_\zeta(\hat{H}, H^*, D) > 0$. We will now show that $d_{\mathrm{TV}}(\hat{H}, H^*) \leq (2 + \zeta)\alpha$, which directly implies that $d_{\mathrm{TV}}(\hat{H}, P) \leq (3 + \zeta)\alpha$ by the triangle inequality. Suppose to the contrary that $d_{\mathrm{TV}}(\hat{H}, H^*) > (2 + \zeta)\alpha$. Then by the definition of $\Gamma_\zeta$, $\hat{P}(\hat{\mathcal{W}}) > H^*(\hat{\mathcal{W}}) + (1 + \zeta/2)\alpha$, where $\hat{\mathcal{W}} = \{x \in \mathcal{X} \mid \hat{H}(x) > H^*(x)\}$. Since $|P(\hat{\mathcal{W}}) - \hat{P}(\hat{\mathcal{W}})| \leq \zeta\alpha/4$, we have $P(\hat{\mathcal{W}}) > H^*(\hat{\mathcal{W}}) + (1 + \zeta/4)\alpha$, which is a contradiction to the assumption that $d_{\mathrm{TV}}(P, H^*) \leq \alpha$. □

While Theorem 1 requires an upper bound $\alpha$ on the accuracy of the best hypothesis, the following theorem obviates this need, at the cost of a mild increase in the sample complexity and a constant factor in the accuracy of the output hypothesis.

**Theorem 3.** *Let $\alpha, \beta, \varepsilon \in (0, 1)$, and $\zeta > 0$ be a constant. Let $\mathcal{H}$ be a set of $m$ distributions and let $P$ be a distribution with $d_{\mathrm{TV}}(P, \mathcal{H}) = \mathrm{OPT}$. There is an $\varepsilon$-differentially private algorithm which takes as input $n$ samples from $P$ and with probability at least $1 - \beta$, outputs a distribution $\hat{H} \in \mathcal{H}$ with $d_{\mathrm{TV}}(P, \hat{H}) \leq 18(3 + \zeta)\mathrm{OPT} + \alpha$, as long as $n \geq O\left(\frac{\log(m/\beta) + \log\log(1/\alpha)}{\alpha^2} + \frac{\log m + \log^2(1/\alpha) \cdot (\log(1/\beta) + \log\log(1/\alpha))}{\alpha\varepsilon}\right)$.*

## 4   An Advanced Method for Private Hypothesis Selection

In Section 3, we provided a simple algorithm whose sample complexity grows logarithmically in the size of the hypothesis class. We now demonstate that this dependence can be improved and, indeed, we can handle infinite hypothesis classes given that their VC dimension is finite and that the cover has small doubling dimension.

To obtain this improved dependence on the hypothesis class size, we must make two improvements to the analysis and algorithm. First, rather than applying a union bound over all the pairwise contests to analyse the tournament, we use a uniform convergence bound in terms of the VC dimension of the Scheffé sets. Second, rather than use the exponential mechanism to select a hypothesis, we use a "GAP-MAX" algorithm [8]. This takes advantage of the fact that, in many cases, even for infinite hypothesis classes, only a handful of hypotheses will have high scores. The GAP-MAX algorithm need only pay for the hypotheses that are close to optimal. To exploit this, we must move to a relaxation of pure differential privacy which is not subject to strong packing lower bounds (as we describe in the supplement). Specifically, we consider approximate differential privacy, although results with an improved dependence are also possible under various variants of concentrated differential privacy [27, 11, 38, 8].

**Theorem 4.** *Let $\mathcal{H}$ be a set of probability distributions on $\mathcal{X}$. Let $d$ be the VC dimension of the set of functions $f_{H,H'} : \mathcal{X} \to \{0, 1\}$ defined by $f_{H,H'}(x) = 1 \iff H(x) > H'(x)$ where $H, H' \in \mathcal{H}$. There exists a $(\varepsilon, \delta)$-differentially private algorithm which has following guarantee. Let $D = \{X_1, \ldots, X_n\}$ be a set of private samples drawn independently from an unknown probability distribution $P$. Let $k = |\{H \in \mathcal{H} : d_{\mathrm{TV}}(H, P) \leq 7\alpha\}|$. Suppose there exists a distribution $H^* \in \mathcal{H}$ such that $d_{\mathrm{TV}}(P, H^*) \leq \alpha$. If $n = \Omega\left(\frac{d + \log(1/\beta)}{\alpha^2} + \frac{\log(k/\beta) + \min\{\log|\mathcal{H}|, \log(1/\delta)\}}{\alpha\varepsilon}\right)$, then the algorithm will output a distribution $\hat{H} \in \mathcal{H}$ such that $d_{\mathrm{TV}}(P, \hat{H}) \leq 7\alpha$ with probability at least*

$1 - \beta$. Alternatively, we can demand that the algorithm be $\frac{1}{2}\varepsilon^2$-concentrated differentially private if $n = \Omega\left(\frac{d + \log(1/\beta)}{\alpha^2} + \frac{\log(k/\beta) + \sqrt{\log|\mathcal{H}|}}{\alpha\varepsilon}\right)$.

Comparing Theorem 4 to Theorem 1, we see that the first (non-private) $\log|\mathcal{H}|$ term is replaced by the VC dimension $d$ and the second (private) $\log|\mathcal{H}|$ term is replaced by $\log k + \log(1/\delta)$. Here $k$ is a measure of the "local" size of the hypothesis class $\mathcal{H}$; its definition is similar to that of the doubling dimension of the hypothesis class under total variation distance. We note that the $\log(1/\delta)$ term could be large, as the privacy failure probability $\delta$ should be cryptographically small. Thus our result includes statements for pure differential privacy (by using the other term in the minimum with $\delta = 0$) and also concentrated differential privacy. Note that, since $d$ and $\log k$ can be upper-bounded by $O(\log|\mathcal{H}|)$, this result supercedes the guarantees of Theorem 1.Further details and a proof of correctness appear in the supplementary material.

## 5 Applications of Hypothesis Selection

In this section, we give a number of applications of Theorem 1, primarily to obtain sample complexity bounds for learning a number of distribution classes of interest. Recall Corollary 1, which is an immediate corollary of Theorem 1. This indicates that we can privately semi-agnostically learn a class of distributions with a number of samples proportional to the log of its covering number.

We instantiate this corollary to give the sample complexity results for semi-agnostically learning product distributions (Section 5.1), Gaussian distributions (Section 5.2), and mixtures (Section 5.3). Additional applications and all proofs of correctness appear in the supplement.

### 5.1 Product Distributions

As a first application, we give an $\varepsilon$-differentially private algorithm for learning product distributions.

**Definition 5.** A $(k, d)$-product distribution *is a distribution over* $[k]^d$, *such that its marginal distributions are independent (i.e., the distribution is the product of its marginals).*

We start by constructing a cover for product distributions.

**Lemma 4.** *There exists an* $\alpha$-*cover of the set of* $(k, d)$-*product distributions of size* $O\left(\frac{kd}{\alpha}\right)^{d(k-1)}$.

With this cover in hand, applying Corollary 1 allows us to conclude the following sample complexity upper bound.

**Corollary 2.** *Suppose we are given a set of samples* $X_1, \ldots, X_n \sim P$, *where* $P$ *is* $\alpha$-*close to a* $(k, d)$-*product distribution. Then for any constant* $\zeta > 0$, *there exists an* $\varepsilon$-*differentially private algorithm which outputs a* $(k, d)$-*product distribution* $H^*$ *such that* $d_{\mathrm{TV}}(P, H^*) \leq (6 + 2\zeta)\alpha$ *with probability* $\geq 9/10$, *so long as* $n = \Omega\left(kd \log\left(\frac{kd}{\alpha}\right)\left(\frac{1}{\alpha^2} + \frac{1}{\alpha\varepsilon}\right)\right)$.

This gives the first $\tilde{O}(d)$ sample algorithm for learning a binary product distribution in under pure differential privacy, improving upon the work of Kamath, Li, Singhal, and Ullman [33] by strengthening the privacy guarantee at a minimal cost in the sample complexity. The natural way to adapt their result from concentrated to pure differential privacy would require $\Omega(d^{3/2})$ samples.

**Remark 1.** *Properly learning a product distribution over* $\{0, 1\}^d$ *to total variation distance* $\leq \frac{1}{2}$ *implies learning its mean* $\mu \in [0, 1]^d$ *up to* $\ell_1$ *error* $\leq 2\sqrt{d}$. *Thus Corollary 2 implies a* $\varepsilon$-*differentially private algorithm which takes* $n = \tilde{O}(d/\varepsilon)$ *samples from a product distribution* $P$ *on* $\{0, 1\}^d$ *and, with high probability, outputs an estimate* $\hat{\mu}$ *of its mean* $\mu$ *with* $\|\hat{\mu} - \mu\|_1 \leq 2\sqrt{d}$. *In contrast, for non-product distributions over the hypercube, estimating the mean to the same accuracy under* $\varepsilon$-*differential privacy requires* $n = \Omega(d^{3/2}/\varepsilon)$ *samples [30, 41]. Thus we have a polynomial separation between estimating product and non-product distributions under pure differential privacy.*

### 5.2 Gaussian Distributions

We next give private algorithms for learning Gaussian distributions. We discuss covers for Gaussian distributions with known and unknown covariance.

**Lemma 5.** *There exists an $\alpha$-cover of the set of Gaussian distributions $\mathcal{N}(\mu, \Sigma)$ in $d$-dimensions with $\|\mu\|_2 \leq R$ and $I \preceq \Sigma \preceq \kappa I$ of size $O\left(\frac{dR}{\alpha}\right)^d \cdot O\left(\frac{d\kappa}{\alpha}\right)^{d(d+1)/2}$. If $\Sigma = I$, the size is $O\left(\frac{dR}{\alpha}\right)^d$.*

In addition, we can obtain bounds of the VC dimension of the Scheffé sets of Gaussian distributions.

**Lemma 6** ([5]). *The set of Gaussian distributions with fixed variance – i.e., all $\mathcal{N}(\mu, I)$ with $\mu \in \mathbb{R}^d$ – has VC dimension $d + 1$. Furthermore, the set of Gaussians with unknown variance – i.e., all $\mathcal{N}(\mu, \Sigma)$ with $\mu \in \mathbb{R}^d$ and $\Sigma \in \mathbb{R}^{d \times d}$ positive definite – has VC dimension $O(d^2)$.*

Combining the covers of Lemma 5 and the VC bound of Lemma 6 with Theorem 4 implies the following corollaries for Gaussian estimation.

**Corollary 3.** *Suppose we are given a set of samples $X_1, \ldots, X_n \sim P$, where $P$ is $\alpha$-close to a Gaussian distribution $\mathcal{N}(\mu, \Sigma)$ in $d$-dimensions with $\|\mu\| \leq R$. Then for any constant $\zeta > 0$, there exists an $\varepsilon$-differentially private algorithm which outputs a Gaussian distribution $H^*$ such that $d_{\mathrm{TV}}(P, H^*) \leq (6+2\zeta)\alpha$ with probability $\geq 9/10$. If $\Sigma = I$, the algorithm requires that $n = \Omega\left(\frac{d}{\alpha^2} + \frac{d}{\alpha\varepsilon}\log\left(\frac{dR}{\alpha}\right)\right)$. If $I \preceq \Sigma \preceq \kappa I$, it requires that $n = \Omega\left(\frac{d^2}{\alpha^2} + \frac{1}{\alpha\varepsilon}\left(d\log\left(\frac{dR}{\alpha}\right) + d^2\log\left(\frac{d\kappa}{\alpha}\right)\right)\right)$.*

Similar to the product distribution case, these are the first $\tilde{O}(d)$ and $\tilde{O}(d^2)$ sample algorithms for learning Gaussians total variation distance under pure differential privacy, improving upon the concentrated differential privacy results of Kamath, Li, Singhal, and Ullman [33].

### 5.2.1 Gaussians with Unbounded Mean

Extending Corollary 3, we consider multivariate Gaussian hypotheses with known covariance and unknown mean, *without* a bound on the mean (the parameter $R$ above). This requires relaxation to approximate differential privacy. In place of Lemma 5, we construct a locally small cover:

**Lemma 7.** *For any $d \in \mathbb{N}$ and $\alpha \in (0, 1/30]$, there exists an $\alpha$-cover $\mathcal{C}_\alpha$ of the set of $d$-dimensional Gaussian distributions $\mathcal{N}(\mu, I)$ satisfying $\forall \mu \in \mathbb{R}^d$ $|\{H \in \mathcal{C}_\alpha : d_{\mathrm{TV}}(H, \mathcal{N}(\mu, I)) \leq 7\alpha\}| \leq 2^{15d}$.*

Applying Theorem 4 with the cover of Lemma 7 and the VC bound from Lemma 6 gives:

**Corollary 4.** *Suppose we are given a set of samples $X_1, \ldots, X_n \sim P$, where $P$ is a spherical Gaussian distribution $\mathcal{N}(\mu, I)$ in $d$-dimensions. Then there exists a $(\varepsilon, \delta)$-differentially private algorithm which outputs a spherical Gaussian distribution $H^*$ such that $d_{\mathrm{TV}}(P, H^*) \leq 7\alpha$ with probability $\geq 1 - 2^{-d}$, so long as $n = \Omega\left(\frac{d}{\alpha^2} + \frac{d + \log(1/\delta)}{\alpha\varepsilon}\right)$.*

Karwa and Vadhan [35] give an algorithm for estimating a univariate Gaussian with unbounded mean, which can be applied independently to the $d$ coordinates to get a sample complexity bound of $\tilde{O}\left(\frac{d}{\alpha^2} + \frac{d}{\alpha\varepsilon} + \frac{\sqrt{d}\log^{3/2}(1/\delta)}{\varepsilon}\right)$. Our bound dominates this except for very small values of $\alpha$.

### 5.3 Mixtures

In this section, we show that our results extend to learning mixtures of classes of distributions.

**Definition 6.** *Let $\mathcal{H}$ be some set of distributions. A $k$-mixture of $\mathcal{H}$ is a distribution with density $\sum_{i=1}^{k} w_i P_i$, where each $P_i \in \mathcal{H}$.*

Our results follow roughly due to the fact that a cover for $k$-mixtures of a class can be written as the Cartesian product of $k$ covers for the class, and then an application of Corollary 1.

**Lemma 8.** *Consider the class of $k$-mixtures of $\mathcal{H}$, where $\mathcal{H}$ is some set of distributions. There exists a $2\alpha$-cover of this class of size $|\mathcal{C}_\alpha|^k \left(\frac{k}{2\alpha} + 1\right)^{k-1}$, where $\mathcal{C}_\alpha$ is an $\alpha$-cover of $\mathcal{H}$.*

**Corollary 5.** *Let $X_1, \ldots, X_n \sim P$, where $P$ is $\alpha$-close to a $k$-mixture of distributions from some set $\mathcal{H}$. Let $\mathcal{C}_\alpha$ be an $\alpha$-cover of the set $\mathcal{H}$, and $\zeta > 0$ be a constant. There exists an $\varepsilon$-differentially private algorithm which outputs a distribution which is $(9+3\zeta)\alpha$-close to $P$ with probability $\geq 9/10$, as long as $n = \Omega\left((k\log|\mathcal{C}_\alpha| + k\log(k/\alpha))\left(\frac{1}{\alpha^2} + \frac{1}{\alpha\varepsilon}\right)\right)$.*

For example, instantiating this for mixtures of Gaussians (and disregarding terms which depend on $R$ and $\kappa$), we get an algorithm with sample complexity $\tilde{O}\left(\frac{kd^2}{\alpha^2} + \frac{kd^2}{\alpha\varepsilon}\right)$.

## Acknowledgments

The authors would like to thank Shay Moran for bringing to their attention the application to PAC learning mentioned in the supplement, Jonathan Ullman for asking questions which motivated Remark 1, and Clément Canonne for assistance in reducing the constant factor in the approximation guarantee. This work was done while the authors were all affiliated the Simons Institute for the Theory of Computing. MB was supported by a Google Research Fellowship, as part of the Simons-Berkeley Research Fellowship program. GK was supported by a Microsoft Research Fellowship, as part of the Simons-Berkeley Research Fellowship program, and the work was also partially done while visiting Microsoft Research, Redmond. TS was supported by a Patrick J. McGovern Research Fellowship, as part of the Simons-Berkeley Research Fellowship program. ZSW was supported in part by a Google Faculty Research Award, a J.P. Morgan Faculty Award, and a Facebook Research Award.

## Footnotes

[1]Roughly, this is due to the fact that the Laplace and Gaussian mechanism are based on $\ell_1$ and $\ell_2$ sensitivity, respectively, and that there is a $\sqrt{d}$-factor relationship between these two norms, in the worst case.

[2]For simplicity of our exposition, we will assume that we can evaluate the two quantities $p_1$ and $p_2$ exactly. In general, we can estimate these quantities to arbitrary accuracy, as long as, for each hypothesis $H$, we can evaluate the density of each point under $H$ and also draw samples from $H$.

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
