[Supplementary Material]

# Private Hypothesis Selection

Mark Bun[*]     Gautam Kamath[†]     Thomas Steinke[‡]     Zhiwei Steven Wu[§]

October 27, 2019

### Abstract

We provide a differentially private algorithm for hypothesis selection. Given samples from an unknown probability distribution $P$ and a set of $m$ probability distributions $\mathcal{H}$, the goal is to output, in a $\varepsilon$-differentially private manner, a distribution from $\mathcal{H}$ whose total variation distance to $P$ is comparable to that of the best such distribution (which we denote by $\alpha$). The sample complexity of our basic algorithm is $O\left(\frac{\log m}{\alpha^2} + \frac{\log m}{\alpha\varepsilon}\right)$, representing a minimal cost for privacy when compared to the non-private algorithm. We also can handle infinite hypothesis classes $\mathcal{H}$ by relaxing to $(\varepsilon, \delta)$-differential privacy.

We apply our hypothesis selection algorithm to give learning algorithms for a number of natural distribution classes, including Gaussians, product distributions, sums of independent random variables, piecewise polynomials, and mixture classes. Our hypothesis selection procedure allows us to generically convert a cover for a class to a learning algorithm, complementing known learning lower bounds which are in terms of the size of the packing number of the class. As the covering and packing numbers are often closely related, for constant $\alpha$, our algorithms achieve the optimal sample complexity for many classes of interest. Finally, we describe an application to private distribution-free PAC learning.

## 1   Introduction

We consider the problem of *hypothesis selection*: given samples from an unknown probability distribution, select a distribution from some fixed set of candidates which is "close" to the unknown distribution in some appropriate distance measure. Such situations can arise naturally in a number of settings. For instance, we may have a number of different methods which work under various circumstances, which are not known in advance. One option is to run all the methods to generate a set of hypotheses, and pick the best from this set afterwards. Relatedly, an algorithm may branch its behavior based on a number of "guesses," which will similarly result in a set of candidates, corresponding to the output at the end of each branch. Finally, if we know that the underlying distribution belongs to some (parametric) class, it is possible to essentially enumerate the class

---

[*]Simons Institute for the Theory of Computing and Boston University. `mbun@bu.edu`. Supported by a Google Research Fellowship, as part of the Simons-Berkeley Research Fellowship program.

[†]Simons Institute for the Theory of Computing and University of Waterloo. `g@csail.mit.edu`. Supported as a Microsoft Research Fellow, as part of the Simons-Berkeley Research Fellowship program. Part of this work was completed while visiting Microsoft Research, Redmond.

[‡]IBM Research. `phs@thomas-steinke.net`. Part of this work completed while visiting the Simons Institute for the Theory of Computing at UC Berkeley.

[§]University of Minnesota, Twin Cities. `zsw@umn.edu`. Part of this work completed while visiting

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

. (We also give an extension of this algorithm which gives guarantees in the *semi-agnostic* learning model; see Section 3.3 for details.) The requirements for this theorem to apply are minimal, and thus it generically provides learning algorithms for a wide variety of hypothesis classes. That said, in non-private settings, the sample complexity given by this method is rather lossy: as an extreme example, there is no finite-size cover of univariate Gaussian distributions with unbounded parameters, so this approach does not give a finite-sample algorithm. That said, it is well-known that $O(1/\alpha^2)$ samples suffice to estimate a Gaussian in total variation distance. In the private setting, our theorem incurs a cost which is somewhat necessary: in particular, it is folklore that any pure $\varepsilon$-differentially private learning algorithm must pay a cost which is logarithmic in the packing number of the class (for completeness, see Lemma 5.1). Due to the relationship between packing and covering numbers (Lemma 5.2), this implies that up to a constant factor relaxation in the learning accuracy, our results are tight (Theorem 5.3). Further discussion appears in Sections 5.

Given Corollary 1.2, in Section 6, we derive new learning results for a number of classes. Our main applications are for $d$-dimensional Gaussian and product distributions. Informally, we obtain $\tilde{O}(d)$ sample algorithms for learning a product distribution and a Gaussian with known covariance (Corollaries 6.3 and 6.10), and an $\tilde{O}(d^2)$ algorithm for learning a Gaussian with unknown covariance (Corollary 6.11). These improve on recent results by Kamath, Li, Singhal, and Ullman [KLSU19] in two different ways. First, as mentioned before, our results are semi-agnostic, so we can handle when the distribution is only *close* to a product or Gaussian distribution. Second, our results hold for pure $(\varepsilon, 0)$-differential privacy, which is a stronger notion than $\varepsilon^2$-zCDP as considered in [KLSU19]. In this weaker model, they also obtained $\tilde{O}(d)$ and $\tilde{O}(d^2)$ sample algorithms, but the natural modifications to achieve $\varepsilon$-DP incur extra poly$(d)$ factors.[1] [KLSU19] also showed $\tilde{\Omega}(d)$

lower bounds for Gaussian and product distribution estimation in the even weaker model of $(\varepsilon, \delta)$-differential privacy. Thus, our results show that the dimension dependence for these problems is unchanged for essentially any notion of differential privacy. In particular, our results show a previously-unknown separation between mean estimation of product distributions and non-product distributions under pure $(\varepsilon, 0)$-differential privacy; see Remark 6.4.

We also apply Theorem 4.1 to obtain algorithms for learning Gaussians under $(\varepsilon, \delta)$-differential privacy, with no bounds on the mean and variance parameters. More specifically, we provide algorithms for learning multivariate Gaussians with unknown mean and known covariance (Corollary 6.13), and univariate Gaussians with both unknown mean and variance (Corollary 6.15). For the former problem, we manage to avoid dependences which arise due to the application of advanced composition (similar to Remark 6.4).

To demonstrate the flexibility of our approach, we also give private learning algorithms for sums of independent random variables (Corollaries 6.20 and 6.22) and piecewise polynomials (Corollary 6.29). To the best of our knowledge, the former class of distributions has not been considered in the private setting, and we rely on covering theorems from the non-private literature. Private learning algorithms for the latter class, piecewise polynomials, have been studied by Diakonikolas, Hardt, and Schmidt [DHS15]. They provide sample and time efficient algorithms for histogram distributions (i.e., piecewise constant distributions), and claim similar results for general piecewise polynomials. Their method depends heavily on rather sophisticated algorithms for the non-private version of this problem [ADLS17]. In constrast, we can obtain comparable sample complexity bounds from just the existence of a cover and elementary VC dimension arguments, which we derive in a fairly self-contained manner.

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

Non-privately, there has been a significant amount of work on learning specific classes of distri-

butions. The PAC-style formulation of the problem we consider originated in [KMR$^+$94]. While learning Gaussians and product distributions can be considered folklore at this point, some of the other classes we learn have enjoyed more recent study. For instance, learning sums of independent random variables was recently considered in [DDS12b] toward the problem of learning Poisson Binomial Distributions (PBDs). Since then, there has been additional work on learning PBDs and various generalizations [DKT15, DDKT16, DKS16b, DKS16c, DKS16a, DLS18].

Piecewise polynomials are a highly-expressive class of distributions, and they can be used to approximate a number of other univariate distribution classes, including distributions which are multi-modal, concave, convex, log-concave, monotone hazard rate, Gaussian, Poisson, Binomial, and more. Algorithms for learning such classes are considered in a number of papers, including [DDS12a, CDSS14a, CDSS14b, ADK15, ADLS17].

There has also been a great deal of work on learning mixtures of distribution classes, particularly mixtures of Gaussians. There are many ways the objective of such a problem can be defined, including clustering [Das99, DS00, AK01, VW02, AM05, CR08b, CR08a, KK10, AS12, RV17, HL18, DKS18, KSS18], parameter estimation [KMV10, MV10, BS10, HK13, ABG$^+$14, BCMV14, HP15, GHK15, XHM16, DTZ17, ABDH$^+$18], proper learning [FOS06, FOS08, DK14, SOAJ14, DKK$^+$16, LS17], and improper learning [CDSS14a]. Our work falls into the line on proper learning: the algorithm is given a set of samples from a mixture of Gaussians, and must output a mixture of Gaussians which is close in total variation distance.

## 1.4 Organization

We begin in Section 2 with preliminaries. In Section 3, we give a basic algorithm for private hypothesis selection, via the exponential mechanism. In Section 4, we extend this approach in two ways: by using VC dimension arguments to reduce the sample complexity for sets of hypotheses with additional structure, and combining this with a GAP-MAX algorithm to achieve non-trivial guarantees for infinite hypothesis classes. Section 5 shows that our approach leads to algorithms which essentially match lower bounds for most distribution classes (in the constant $\alpha$ regime). We consider applications in Section 6: through a combination of arguments about covers and VC dimension, we derive algorithms for learning a number of classes of distributions, as well as describe an application to private PAC learning. Finally, we conclude in Section 7 with open questions.

# 2 Preliminaries

We start with some preliminaries and definitions.

**Definition 2.1.** *The* total variation distance *or* statistical distance *between $P$ and $Q$ is defined as*

$$d_{\mathrm{TV}}(P,Q) = \max_{S \subseteq \Omega} P(S) - Q(S) = \frac{1}{2} \int_{x \in \Omega} |P(x) - Q(x)| dx = \frac{1}{2}\|P - Q\|_1 \in [0,1].$$

*Moreover, if $\mathcal{H}$ is a set of distributions over a common domain, we define $d_{\mathrm{TV}}(P,\mathcal{H}) = \inf_{H \in \mathcal{H}} d_{\mathrm{TV}}(P,H)$.*

Throughout this paper, we consider packings and coverings of sets of distributions with respect to total variation distance.

**Definition 2.2.** *A $\gamma$-cover of a set of distributions $\mathcal{H}$ is a set of distributions $\mathcal{C}_\gamma$, such that for every $H \in \mathcal{H}$, there exists some $P \in \mathcal{C}_\gamma$ such that $d_{\mathrm{TV}}(P,H) \leq \gamma$.*

*A $\gamma$-packing of a set of distributions $\mathcal{H}$ is a set of distributions $\mathcal{P}_\gamma \subseteq \mathcal{H}$, such that for every pair of distributions $P,Q \in \mathcal{P}_\gamma$, we have that $d_{\mathrm{TV}}(P,Q) \geq \gamma$.*

In this paper, we present semi-agnostic learning algorithms.

**Definition 2.3.** *An algorithm is said to be an $\alpha$-semi-agnostic learner for a class $\mathcal{H}$ if it has the following guarantees. Suppose we are given $X_1, \ldots, X_n \sim P$, where $d_{\mathrm{TV}}(P, \mathcal{H}) \leq \mathrm{OPT}$. The algorithm must output some distribution $\hat{H}$ such that $d_{\mathrm{TV}}(P, H) \leq c \cdot \mathrm{OPT} + O(\alpha)$, for some constant $c \geq 1$. If $c = 1$, then the algorithm is said to be agnostic.*

Now we define differential privacy. We say that $D$ and $D'$ are neighboring datasets, denoted $D \sim D'$, if $D$ and $D'$ differ by at most one observation. Informally, differential privacy requires that the algorithm has close output distributions when run on any pair of neighboring datasets. More formally:

**Definition 2.4** ([DMNS06]). *A randomized algorithm $T : X^* \to \mathcal{R}$ is $(\varepsilon, \delta)$-differentially private if for all $n \geq 1$, for all neighboring datasets $D, D' \in X^n$, and for all events $S \subseteq \mathcal{R}$,*

$$\Pr[T(D) \in S] \leq e^{\varepsilon} \Pr[T(D') \in S] + \delta.$$

*If $\delta = 0$, we say that $T$ is $\varepsilon$-differentially private.*

We will also use the related notion of concentrated differential privacy:

**Definition 2.5** ([DR16, BS16]). *A randomized algorithm $T : X^* \to \mathcal{R}$ satisfies $\rho$-zero-concentrated differential privacy if for all $n \geq 1$, for all neighboring datasets $D, D' \in X^n$, and for all $\alpha \in (1, \infty)$,*

$$R_{\alpha}(M(D) \| M(D')) \leq \rho\alpha,$$

*where $R_{\alpha}(M(D) \| M(D'))$ is the $\alpha$-Rényi divergence between $M(D)$ and $M(D')$.*[2]

The exponential mechanism [MT07] is a powerful $\varepsilon$-differentially private mechanism for selecting an approximately best outcome from a set of alternatives, where the quality of an outcome is measured by a score function relating each alternative to the underlying dataset. Letting $\mathcal{R}$ be the set of possible outcomes, a score function $q : X^* \times \mathcal{R} \to \mathbb{R}$ maps each pair consisting of a dataset and an outcome to a real-valued score. The exponential mechanism $\mathcal{M}_E$ instantiated with a dataset $D$, a score function $q$, and a privacy parameter $\varepsilon$ selects an outcome $r$ in $\mathcal{R}$ with probability proportional to $\exp\left(\varepsilon q(D, r)/(2\Delta(q))\right)$, where $\Delta(q)$ is the sensitivity of the score function defined as

$$\Delta(q) = \max_{r \in \mathcal{R}, D \sim D'} \left| q(D, r) - q(D', r) \right|.$$

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

## 3.3 Obtaining a Semi-Agnostic Algorithm

Theorem 1.1 shows that given a hypothesis class $\mathcal{H}$ and samples from an unknown distribution $P$, we can privately find a distribution $\hat{H} \in \mathcal{H}$ with $d_{\mathrm{TV}}(P, \hat{H}) \leq (3 + \zeta)\alpha$ *provided* that we know $d_{\mathrm{TV}}(P, \mathcal{H}) \leq \alpha$. But what if we are not promised that $P$ is itself close to $\mathcal{H}$? We would like to design a private hypothesis selection algorithm for the more general semi-agnostic setting, where for any value of $\mathrm{OPT} := d_{\mathrm{TV}}(P, \mathcal{H})$, we are able to privately identify a distribution $\hat{H} \in \mathcal{H}$ with $d_{\mathrm{TV}}(P, \hat{H}) \leq c \cdot \mathrm{OPT} + \alpha$ for some universal constant $c$. Our goal will be to do this with sample complexity which is still logarithmic in $|\mathcal{H}|$.

Our strategy for handling this more general setting is by a reduction to that of Theorem 1.1. We run that algorithm $T = O(\log(1/\alpha))$ times, doubling the choice of $\alpha$ in each run and producing a sequence of candidate hypotheses $H_1, \ldots, H_T$. By the guarantees of Theorem 1.1, there is some candidate $H_t$ with $d_{\mathrm{TV}}(P, H_t) \leq 2(3 + \zeta)\mathrm{OPT}$. The remaining task is to approximately select the best candidate from $H_1, \ldots, H_T$. This is done by implementing a private version of the Scheffé tournament which is itself semi-agnostic, but has a very poor (quadratic) dependence on the number of candidates $T$.

We prove the following result, which gives a semi-agnostic learner whose sample complexity is comparable to that of Theorem 1.1.

**Theorem 3.4.** *Let $\alpha, \beta, \varepsilon \in (0, 1)$, and $\zeta > 0$ be a constant. Let $\mathcal{H}$ be a set of $m$ distributions and let $P$ be a distribution with $d_{\mathrm{TV}}(P, \mathcal{H}) = \mathrm{OPT}$. There is an $\varepsilon$-differentially private algorithm which takes as input $n$ samples from $P$ and with probability at least $1 - \beta$, outputs a distribution $\hat{H} \in \mathcal{H}$ with $d_{\mathrm{TV}}(P, \hat{H}) \leq 18(3 + \zeta)\mathrm{OPT} + \alpha$, as long as*

$$n \geq O\left(\frac{\log(m/\beta) + \log\log(1/\alpha)}{\alpha^2} + \frac{\log m + \log^2(1/\alpha) \cdot (\log(1/\beta) + \log\log(1/\alpha))}{\alpha\varepsilon}\right).$$

As discussed above, the algorithm relies on the following variant with a much worse dependence on $m$.

**Lemma 3.5.** *Let $\alpha, \beta, \varepsilon \in (0, 1)$. There is an $\varepsilon$-differentially private algorithm which takes as input $n$ samples from $P$ and with probability at least $1 - \beta$, outputs a distribution $\hat{H} \in \mathcal{H}$ with $d_{\mathrm{TV}}(P, \hat{H}) \leq 9\,\mathrm{OPT} + \alpha$, as long as*

$$n \geq O\left(\frac{\log(m/\beta)}{\alpha^2} + \frac{m^2 \log(m/\beta)}{\alpha\varepsilon}\right).$$

*Proof sketch.* We use a different variation of the Scheffé tournament which appears in [DL01]. Non-privately, the algorithm works as follows. For every pair of hypotheses $H, H' \in \mathcal{H}$ with Scheffé set $\mathcal{W}_{H,H'} = \{x \in \mathcal{X} \mid H(x) > H'(x)\}$, let $H(\mathcal{W}_{H,H'})$, $H'(\mathcal{W}_{H,H'})$, and $P(\mathcal{W}_{H,H'})$ denote the probability masses of $H, H', P$ on $\mathcal{W}_{H,H'}$, respectively. Moreover, let $\hat{P}(\mathcal{W}_{H,H'})$ denote the fraction of points in the input sample $D$ which lie in $\mathcal{W}_{H,H'}$. We declare $H$ to be the winner of the pairwise contest between $H$ and $H'$ if $|H(\mathcal{W}_{H,H'}) - \hat{P}(\mathcal{W}_{H,H'})| < |H'(\mathcal{W}_{H,H'}) - \hat{P}(\mathcal{W}_{H,H'})|$. Otherwise, we declare $H'$ to be the winner. The algorithm outputs the hypothesis $\hat{H}$ which wins the most pairwise contests (breaking ties arbitrarily).

To make this algorithm $\varepsilon$-differentially private, we replace $\hat{P}(\mathcal{W}_{H,H'})$ in each pairwise contest with the $(\varepsilon/\binom{m}{2})$-differentially private estimate $c_{H,H'} = \hat{P}(\mathcal{W}_{H,H'}) + \mathrm{Lap}(\binom{m}{2}/\varepsilon n)$. By the composition guarantees of differential privacy, the algorithm as a whole is $\varepsilon$-differentially private.

The analysis of Devroye and Lugosi [DL01, Theorem 6.2] shows that the (private) Scheffé tournament outputs a hypothesis $\hat{H}$ with

$$d_{\mathrm{TV}}(\hat{H}, P) \leq 9\,\mathrm{OPT} + 16 \max_{H,H' \in \mathcal{H}} \left| P(\mathcal{W}_{H,H'}) - c_{H,H'} \right|.$$

Fix an arbitrary pair $H, H'$. A Chernoff bound shows that $|P(\mathcal{W}_{H,H'}) - \hat{P}(\mathcal{W}_{H,H'})| \le \alpha/32$ with probability at least $1 - \beta/(2m^2)$ as long as $n \ge O(\ln(m/\beta)/\alpha^2)$. Moreover, properties of the Laplace distribution guarantee $|c_{H,H'} - \hat{P}(\mathcal{W}_{H,H'})| \le \alpha/32$ with probability at least $1 - \beta/(2m^2)$ as long as $n \ge O(m^2 \log(m/\beta)/\alpha\varepsilon)$. The triangle inequality and a union bound over all pairs $H, H'$ complete the proof. $\square$

*Proof of Theorem 3.4.* We now combine the private hypothesis selection algorithm of Theorem 1.1 with the expensive semi-agnostic learner of Lemma 3.5 to prove Theorem 3.4. Define sequences $\alpha_1 = \alpha/126, \alpha_2 = 2\alpha/126, \ldots, \alpha_T = 2^{T-1}\alpha/126$ and $\varepsilon_1 = \varepsilon/4, \varepsilon_2 = \varepsilon/8, \ldots, \varepsilon_T = 2^{-(T+1)}\varepsilon$ for $T = \lceil \log_2(1/\alpha) \rceil + 1$. For each $t = 1, \ldots, T$, let $H_t$ denote the outcome of a run of Algorithm 2 using accuracy parameter $\alpha_t$ and privacy parameter $\varepsilon_t$. Finally, use the algorithm of Lemma 3.5 to select a hypothesis from $H_0, \ldots, H_T$ using accuracy parameter $\alpha$ and privacy parameter $\varepsilon/2$.

Privacy of this algorithm follows immediately from composition of differential privacy. We now analyze its sample complexity guarantee. By Lemma 3.3, we have that all $T$ runs of Algorithm 2 succeed simultaneously with probability at least $1 - \beta/2$ as long as

$$n \ge O\left(\frac{\log(m/\beta) + \log\log(1/\alpha)}{\alpha^2} + \frac{\log(m/\beta) + \log\log(1/\alpha)}{\alpha\varepsilon}\right).$$

Condition on this event occurring. Recall that success of run $t$ of Algorithm 2 means that if $\mathrm{OPT} \in (\alpha_{t-1}, \alpha_t]$, then $d_{\mathrm{TV}}(P, H_t) \le (3 + \zeta)\alpha_t \le 2(3 + \zeta)\,\mathrm{OPT}$. Meanwhile, if $\mathrm{OPT} \le \alpha_1 = \alpha/126$, then we have $d_{\mathrm{TV}}(P, H_1) \le \alpha/18$. Hence, regardless of the value of $\mathrm{OPT}$, there exists a run $t$ such that $d_{\mathrm{TV}}(P, H_t) \le 2(3 + \zeta)\,\mathrm{OPT} + \alpha/18$. The algorithm of Lemma 3.5 is now, with probability at least $1 - \beta/2$, able to select a hypothesis $\hat{H}$ with $d_{\mathrm{TV}}(P, \hat{H}) \le 9d_{\mathrm{TV}}(P, H_t) + \alpha/2 \le 18(3 + \zeta)\,\mathrm{OPT} + \alpha$ as long as

$$n \ge O\left(\frac{\log(1/\beta) + \log\log(1/\alpha)}{\alpha^2} + \frac{\log^2(1/\alpha) \cdot (\log(1/\beta) + \log\log(1/\alpha))}{\alpha\varepsilon}\right).$$

This gives the asserted sample complexity guarantee. $\square$

## 4   An Advanced Method for Private Hypothesis Selection

In Section 3, we provided a simple algorithm whose sample complexity grows logarithmically in the size of the hypothesis class. We now demonstate that this dependence can be improved and, indeed, we can handle infinite hypothesis classes given that their VC dimension is finite and that the cover has small doubling dimension.

To obtain this improved dependence on the hypothesis class size, we must make two improvements to the analysis and algorithm. First, rather than applying a union bound over all the pairwise contests to analyse the tournament, we use a uniform convergence bound in terms of the VC dimension of the Scheffé sets. Second, rather than use the exponential mechanism to select a hypothesis, we use a "GAP-MAX" algorithm [BDRS18]. This takes advantage of the fact that, in many cases, even for infinite hypothesis classes, only a handful of hypotheses will have high scores. The GAP-MAX algorithm need only pay for the hypotheses that are close to optimal. To exploit this, we must move to a relaxation of pure differential privacy which is not subject to strong packing lower bounds (as we describe in Section 5). Specifically, we consider approximate differential privacy, although results with an improved dependence are also possible under various variants of concentrated differential privacy [DR16, BS16, Mir17, BDRS18].

**Theorem 4.1.** *Let $\mathcal{H}$ be a set of probability distributions on $\mathcal{X}$. Let $d$ be the VC dimension of the set of functions $f_{H,H'} : \mathcal{X} \to \{0,1\}$ defined by $f_{H,H'}(x) = 1 \iff H(x) > H'(x)$ where $H, H' \in \mathcal{H}$. There exists a $(\varepsilon, \delta)$-differentially private algorithm which has following guarantee. Let $D = \{X_1, \ldots, X_n\}$ be a set of private samples drawn independently from an unknown probability distribution $P$. Let $k = |\{H \in \mathcal{H} : d_{\mathrm{TV}}(H, P) \leq 7\alpha\}|$. Suppose there exists a distribution $H^* \in \mathcal{H}$ such that $d_{\mathrm{TV}}(P, H^*) \leq \alpha$. If $n = \Omega\left(\frac{d + \log(1/\beta)}{\alpha^2} + \frac{\log(k/\beta) + \min\{\log|\mathcal{H}|, \log(1/\delta)\}}{\alpha\varepsilon}\right)$, then the algorithm will output a distribution $\hat{H} \in \mathcal{H}$ such that $d_{\mathrm{TV}}(P, \hat{H}) \leq 7\alpha$ with probability at least $1 - \beta$.*

*Alternatively, we can demand that the algorithm be $\frac{1}{2}\varepsilon^2$-concentrated differentially private if $n = \Omega\left(\frac{d + \log(1/\beta)}{\alpha^2} + \frac{\log(k/\beta) + \sqrt{\log|\mathcal{H}|}}{\alpha\varepsilon}\right)$.*

Comparing Theorem 4.1 to Theorem 1.1, we see that the first (non-private) $\log|\mathcal{H}|$ term is replaced by the VC dimension $d$ and the second (private) $\log|\mathcal{H}|$ term is replaced by $\log k + \log(1/\delta)$. Here $k$ is a measure of the "local" size of the hypothesis class $\mathcal{H}$; its definition is similar to that of the doubling dimension of the hypothesis class under total variation distance.

We note that the $\log(1/\delta)$ term could be large, as the privacy failure probability $\delta$ should be cryptographically small. Thus our result includes statements for pure differential privacy (by using the other term in the minimum with $\delta = 0$) and also concentrated differential privacy. Note that, since $d$ and $\log k$ can be upper-bounded by $O(\log|\mathcal{H}|)$, this result supercedes the guarantees of Theorem 1.1.

## 4.1 VC Dimension

We begin by reviewing the definition of Vapnik-Chervonenkis (VC) dimension and its properties.

**Definition 4.2** (VC dimension [VC74])**.** *Let $\mathcal{F}$ be a set of functions $f : \mathcal{X} \to \{0,1\}$. The VC dimension of $\mathcal{F}$ is defined to be the largest $d$ such that there exist $x_1, \cdots, x_d \in \mathcal{X}$ and $f_1, \cdots, f_{2^d} \in \mathcal{H}$ such that for all $1 \leq i < j \leq 2^d$ there exists $1 \leq k \leq d$ such that $f_i(x_k) \neq f_j(x_k)$.*

For our setting, we must extend the definition of VC dimension from function families to hypothesis classes.

**Definition 4.3** (VC dimension of hypothesis class)**.** *Let $\mathcal{H}$ be a set of probability distributions on a space $\mathcal{X}$. For $H, H' \in \mathcal{H}$, define $f_{H,H'} : \mathcal{X} \to \{0,1\}$ by $f(x) = 1 \iff H(x) > H'(x)$. Define $\mathcal{F}(\mathcal{H}) = \{f_{H,H'} : H, H' \in \mathcal{H}\}$. We define the VC dimension of $\mathcal{H}$ to be the VC dimension of $\mathcal{F}(\mathcal{H})$.*[4]

The key property of VC dimension is the following uniform convergence bound, which we use in place of a union bound.

**Theorem 4.4** (Uniform Convergence [Tal94])**.** *Let $\mathcal{F}$ be a set of functions $f : \mathcal{X} \to \{0,1\}$ with VC dimension $d$. Let $P$ be a distribution on $\mathcal{X}$. Then*

$$\Pr_{D \leftarrow P^n}\left[\sup_{f \in \mathcal{F}} |f(D) - f(P)| \leq \alpha\right] \geq 1 - \beta$$

*whenever $n = \Omega\left(\frac{d + \log(1/\beta)}{\alpha^2}\right)$. Here $f(D) := \frac{1}{n}\sum_{x \in D} f(x)$ and $f(P) := \mathbf{E}_{X \leftarrow P}[f(X)]$.*

It is immediate from Definition 4.2 that $VC(\mathcal{F}) \leq \lfloor \log_2 |\mathcal{F}| \rfloor$. Thus Theorem 4.4 subsumes the union bound used in the proof of Theorem 1.1.

The relevant application of uniform convergence for our algorithm is the following lemma (roughly the equivalent of Lemma 3.1), which says that good hypotheses have high scores, and bad hypotheses have low scores.

**Lemma 4.5.** *Let $\mathcal{H}$ be a collection of probability distributions on $\mathcal{X}$ with VC dimension $d$. Let $S : \mathcal{H} \times \mathcal{X}^n \to \mathbb{R}$ be as in Equation 1, namely*

$$S(H, D) = \inf_{H' \in \mathcal{H}} \max \left\{ \begin{array}{c} |\{x \in D : H(x) > H'(x)\}| - n \cdot (\Pr_{X \leftarrow H'}[H(X) > H'(X)] + 3\alpha), \\ n \cdot \mathbb{I}[d_{\mathrm{TV}}(H, H') \leq 6\alpha] \end{array} \right\},$$

*where $\mathbb{I}$ denotes the indicator function.*

*Let $P$ be a distribution on $\mathcal{X}$. Let $\alpha, \beta > 0$ and $n \geq O(\frac{1}{\alpha^2}(d + \log(1/\beta)))$. Suppose there exists $H^* \in \mathcal{H}$ with $d_{\mathrm{TV}}(P, H^*) \leq \alpha$. Then, with probability at least $1 - \beta$ over $D \leftarrow P^n$, we have*

- *$S(H^*, D) > \alpha n$ and*

- *$S(H, D) = 0$ for all $H \in \mathcal{H}$ with $d_{\mathrm{TV}}(H, P) > 7\alpha$.*

*Proof.* For $H, H' \in \mathcal{H}$, define $f_{H,H'} : \mathcal{X} \to \{0, 1\}$ by $f_{H,H'}(x) = 1 \iff H(x) > H'(x)$. Note that $|\{x \in D : H(x) > H'(x)\}| = \sum_{x \in D} f_{H,H'}(x)$ and $d$ is the VC dimension of the function class $\{f_{H,H'} : H, H' \in \mathcal{H}\}$. By Theorem 4.4, if $n = \Omega\left(\frac{d + \log(1/\beta)}{\alpha^2}\right)$, then

$$\Pr_{D \leftarrow P^n}\left[\forall H, H' \in \mathcal{H} \quad \left||\{x \in D : H(x) > H'(x)\}| - n \cdot \Pr_{X \leftarrow P}[H(X) > H'(X)]\right| \leq \alpha n\right] \geq 1 - \beta.$$

We condition on this event happening.

In order to prove the first conclusion – namely, $S(H^*, D) > \alpha n$ – it remains to show that, for all $H' \in \mathcal{H}$, we have either $d_{\mathrm{TV}}(H^*, H') \leq 6\alpha$ or

$$|\{x \in D : H(x) > H'(x)\}| - n \cdot (\Pr_{X \leftarrow H'}[H^*(X) > H'(X)] + 3\alpha) > \alpha n.$$

If $d_{\mathrm{TV}}(H^*, H') \leq 6\alpha$, we are done, so assume $d_{\mathrm{TV}}(H^*, H') > 6\alpha$. By the uniform convergence event we have conditioned on,

$$\begin{aligned}
|\{x \in D : H(x) > H'(x)\}| &\geq n \cdot (\Pr_{X \leftarrow P}[H(X) > H'(X)] - \alpha) \\
&\geq n \cdot (\Pr_{X \leftarrow H^*}[H(X) > H'(X)] - d_{\mathrm{TV}}(P, H^*) - \alpha) \\
&\geq n \cdot (d_{\mathrm{TV}}(H^*, H') + \Pr_{X \leftarrow H'}[H(X) > H'(X)] - 2\alpha) \\
&> n \cdot (6\alpha + \Pr_{X \leftarrow H'}[H(X) > H'(X)] - 2\alpha),
\end{aligned}$$

from which the desired conclusion follows.

In order to prove the second conclusion – namely, $S(H, D) = 0$ for all $H \in \mathcal{H}$ with $d_{\mathrm{TV}}(H, P) > 7\alpha$ – it suffices to show that one $H' \in \mathcal{H}$ yields a score of zero for any $H \in \mathcal{H}$ with $d_{\mathrm{TV}}(H, P) > 7\alpha$. In particular, we show that $H' = H^*$ yields a score of zero for any such $H$. That is, if $d_{\mathrm{TV}}(H, P) > 7\alpha$, then $d_{\mathrm{TV}}(H, H^*) > 6\alpha$ and

$$|\{x \in D : H(x) > H^*(x)\}| - n \cdot (\Pr_{X \leftarrow H^*}[H(X) > H^*(X)] + 3\alpha) \leq 0.$$

By the triangle inequality $d_{\mathrm{TV}}(H, H^*) \geq d_{\mathrm{TV}}(H, P) - d_{\mathrm{TV}}(P, H^*) > 7\alpha - \alpha = 6\alpha$, as required. By the uniform convergence event we have conditioned on,

$$\begin{aligned}
|\{x \in D : H(x) > H^*(x)\}| &\leq n \cdot (\Pr_{X \leftarrow P}[H(X) > H^*(X)] + \alpha) \\
&\leq n \cdot (\Pr_{X \leftarrow H^*}[H(X) > H^*(X)] + d_{\mathrm{TV}}(P, H^*) + \alpha) \\
&\leq n \cdot (\Pr_{X \leftarrow H^*}[H(X) > H^*(X)] + 2\alpha),
\end{aligned}$$

which completes the proof. $\square$

## 4.2 GAP-MAX Algorithm

In place of the exponential mechanism for privately selecting a hypothesis we use the following algorithm that works under a "gap" assumption. That is, we assume that there is a $5\alpha n$ gap between the highest score and the $(k+1)$-th highest score. Rather than paying in sample complexity for the total number of hypotheses we pay for the number of high-scoring hypotheses $k$.

This algorithm is based on the GAP-MAX algorithm of Bun, Dwork, Rothblum, and Steinke [BDRS18]. However, we combine their GAP-MAX algorithm with the exponential mechanism to improve the dependence on the parameter $k$.

**Theorem 4.6.** *Let $\mathcal{H}$ and $\mathcal{X}$ be arbitrary sets. Let $S : \mathcal{H} \times \mathcal{X}^n \to \mathbb{R}$ have sensitivity at most 1 in its second argument – that is, for all $H \in \mathcal{H}$ and all $D, D' \in \mathcal{X}^n$ differing in a single example, $|S(H, D) - S(H, D')| \leq 1$.*

*For $D \in \mathcal{X}^n$ and $\alpha > 0$, define*

$$K(D, 5\alpha) := \left| \left\{ H \in \mathcal{H} : S(H, D) \geq \sup_{H' \in \mathcal{H}} S(H', D) - 5\alpha n \right\} \right|.$$

*Given parameters $\varepsilon, \delta, \beta > 0$ and $n, k \geq 1$, there exists a $(\varepsilon, \delta)$-differentially private randomized algorithm $M : \mathcal{X}^n \to \mathcal{H}$ such that, for all $D \in \mathcal{X}^n$ and all $\alpha > 0$,*

$$K(D, 5\alpha) \leq k \implies \Pr\left[ S(M(D), D) \geq \sup_{H' \in \mathcal{H}} S(H', D) - \alpha n \right] \geq 1 - \beta$$

*provided $n = \Omega\left( \frac{\min\{\log |\mathcal{H}|, \log(1/\delta)\} + \log(k/\beta)}{\alpha\varepsilon} \right)$.*

*Furthermore, given $\varepsilon, \beta > 0$ and $n, k \geq 1$, there exists a $\frac{1}{2}\varepsilon^2$-concentrated differentially private [BS16] algorithm $M : \mathcal{X}^n \to \mathcal{H}$ such that, for all $D \in \mathcal{X}^n$ and all $\alpha > 0$,*

$$K(D, 5\alpha) \leq k \implies \Pr\left[ S(M(D), D) \geq \sup_{H' \in \mathcal{H}} S(H', D) - \alpha n \right] \geq 1 - \beta$$

*provided $n = \Omega\left( \frac{\sqrt{\log |\mathcal{H}|} + \log(k/\beta)}{\alpha\varepsilon} \right)$.*

*Proof.* We begin by describing the algorithm.

1. Let $m = \left\lceil \frac{k^2}{\beta} \right\rceil$ and let $G : \mathcal{H} \to [m]$ be a uniformly random function.[5]

2. Randomly select $B \in [m]$ with

$$\Pr[B = b] \propto \exp\left( \frac{\varepsilon}{4} \sup\left\{ S(H, D) : H \in \mathcal{H}, G(H) = b \right\} \right).$$

3. Define $\mathcal{H}_B = \{H \in \mathcal{H} : G(H) = B\}$. Let $H_B^1 = \operatorname{argmax}_{H \in \mathcal{H}_B} S(H, D)$ and $H_B^2 = \operatorname{argmax}_{H \in \mathcal{H}_B \setminus \{H_B^1\}} S(H, D)$, breaking ties arbitrarily. (That is, $\mathcal{H}_B$ is the $B$-th "bin" and $H_B^1$ and $H_B^2$ are the items in this bin with the largest and second-largest scores respectively.) Define $S_B' : \mathcal{H}_B \times \mathcal{X}^n \to \mathbb{R}$ by

$$S_B'(H, D) = \frac{1}{2} \max\{0, S(H, D) - S(H_B^2, D)\}.$$

(Note that $S_B'$ has sensitivity 1 and $S_B'(H, D) = 0$ whenever $H \neq H_B^1$.)

4. Let $\mathcal{D}$ be a distribution on $\mathbb{R}$ such that adding a sample from $\mathcal{D}$ to a sensitivity-1 function provides $(\varepsilon/4, \delta/2)$-differential privacy (or, respectively, $\frac{1}{6}\varepsilon^2$-concentrated differential privacy). For example, $\mathcal{D}$ could be a Laplace distribution with scale $4/\varepsilon$ truncated to the interval $[-t, t]$ for $t = 4(1 + \log(1/\delta))/\varepsilon$ (or unbounded if $\delta = 0$). To attain concentrated differential privacy, we can set $\mathcal{D} = N\left(0, \frac{3}{\varepsilon^2}\right)$, a centered Gaussian with variance $3/\varepsilon^2$.

5. Draw a sample $Z_H$ i.i.d. from $\mathcal{D}$ corresponding to every $H \in \mathcal{H}_B$.

6. Return $H^* = \operatorname{argmax}_{H \in \mathcal{H}_B} S'_B(H, D) + Z_H$.

The selection of $B$ is an instantiation of the exponential mechanism [MT07] and is $(\varepsilon/2, 0)$-differentially private. The selection of $H^*$ in the final step is a GAP-MAX algorithm [BDRS18] and is $(\varepsilon/2, \delta)$-differentially private. By composition, the entire algorithm is $(\varepsilon, \delta)$-differentially private (or, respectively, $\frac{1}{2}\varepsilon^2$-concentrated differentially private).

For the utility analysis, in order for the algorithm to output a good $H^*$, it suffices for the following three events to occur.

- $S(H_B^1, D) \geq \sup_{H' \in \mathcal{H}} S(H', D) - \alpha n$.
  That is, restricting the search to $\mathcal{H}_B$, rather than all of $\mathcal{H}$, only reduces the score of the optimal choice by $\alpha n$. The exponential mechanism ensures that this happens with probability at least $1 - \beta/4$, as long as $n \geq \frac{4 \log(2k/\beta)}{\varepsilon \alpha}$.

- $S(H_B^2, D) < \sup_{H' \in \mathcal{H}} S(H', D) - 5\alpha n$.
  That is, the second-highest score within $\mathcal{H}_B$ is at least $5\alpha n$ less than the highest score overall. We have assumed that there are at most $k$ elements $H \in \mathcal{H}$ such that $S(H, D) \geq \sup_{H' \in \mathcal{H}} S(H', D) - 5\alpha n$. Call these "large elements." Since $G$ is random and $m \geq k^2/\beta$, the probability that more than one large element satisfies $G(H) = B$ is at most $\beta/2$. That is to say, with high probability there are no collisions under the hash function $G$ of the $k$ large elements. This suffices for the event to occur.

- $\sup_{H \in \mathcal{H}_B} |Z_H| \leq \alpha n$.
  If the noise distribution $\mathcal{D}$ is supported on $[-\alpha n, \alpha n]$, then this condition holds with probability 1. For the truncated Laplace distribution, this is possible whenever $n \geq 1 + 4 \log(1/\delta)/\alpha\varepsilon$. Alternatively, we can use unbounded Laplace noise and a union bound to show that this event occurs with probability at least $1 - \beta/4$ whenever $n \geq 4 \log(4|\mathcal{H}_B|/\beta)/\varepsilon\alpha$. For Gaussian noise, $n \geq \frac{3}{\varepsilon\alpha}\sqrt{\log(4|\mathcal{H}_B|/\beta)}$ suffices.

Assuming the first and second events occur, we have $S'_B(H_B^1, D) = \frac{S(H_B^1, D) - S(H_B^2, D)}{2} > 2\alpha n$. Given this, the third event implies $H^* = H_B^1$. Finally, the first event then implies $S(H^*, D) \geq \sup_{H' \in \mathcal{H}} S(H', D) - \alpha n$, as required. A union bound over the three events completes the proof. $\square$

Now we can combine the VC-based uniform convergence bound with the GAP-MAX algorithm to prove our result.

*Proof of Theorem 4.1.* By Lemma 4.5, with high probability over the draw of the dataset $D$, our score function satisfies $\sup_{H \in \mathcal{H}} S(H, D) \geq S(H^*, D) > \alpha n$ and $S(H, D) = 0$ whenever $d_{\mathrm{TV}}(H, P) > 7\alpha$. This requires $n = \Omega(d/\alpha^2)$.

Note that the score function $S$ has sensitivity-1, since it is the supremum of counts. Conditioned on the uniform convergence event, the maximum score is at least $\alpha n$ and there are at most $k$ elements of $\mathcal{H}$ with score greater than 0. Thus we can apply the GAP-MAX algorithm of Theorem 4.6. If $n = \Omega((\min\{\log |\mathcal{H}|, \log(1/\delta)\} + \log(k))/\alpha\varepsilon)$, then with high probability, the algorithm outputs $\hat{H} \in \mathcal{H}$ with score at least $\frac{4}{5}\alpha n$, as required. $\square$

# 5 Packings, Lower Bounds, and Relations to Covers

In this section, we show that the sample complexity of our algorithms for private hypothesis selection with pure differential privacy cannot be improved, at least for constant values of the proximity parameter $\alpha$. We first apply a packing argument [HT10, BBKN14] to show a lower bound which is logarithmic in the packing number of the class of distributions (Lemma 5.1). We then state a folklore relationship between the sizes of maximal packings and minimal covers (Lemma 5.2), which shows that instantiating our private hypothesis selection algorithm with a minimal cover gives essentially optimal sample complexity (Theorem 5.3).

**Lemma 5.1.** *Suppose there exists an $\alpha$-packing $\mathcal{P}_\alpha$ of a set of distributions $\mathcal{H}$. Then any $\varepsilon$-differentially private algorithm which takes as input samples $X_1, \ldots, X_n \sim P$ for some $P \in \mathcal{H}$ and produces a distribution $\hat{H}$ such that $d_{\mathrm{TV}}(P, \hat{H}) \leq \alpha$ with probability $\geq 9/10$ requires*

$$n = \Omega\left(\frac{\log |\mathcal{P}_\alpha|}{\varepsilon}\right).$$

*Proof.* Let $M$ be a $\varepsilon$-differentially private algorithm with the stated accuracy requirement, and denote by $M(P^n)$ the distribution on hypotheses obtained by running $M$ on $n$ i.i.d. samples from a distribution $P \in \mathcal{H}$. For each $P \in \mathcal{P}_\alpha$, let $B_P$ denote the set of distributions which are at total variation distance at most $\alpha$ from $P$. Then the accuracy requirement implies that $\Pr_{\hat{H} \leftarrow M(P^n)}\left[\hat{H} \in B_P\right] \geq 9/10$. Let $P_0 \in \mathcal{P}_\alpha$ be an arbitrary packing element. Then by group privacy applied to groups of size $n$, we have

$$\Pr_{\hat{H} \leftarrow M(P_0^n)}\left[\hat{H} \in B_P\right] \geq e^{-\varepsilon n} \cdot 9/10$$

for every $P \in \mathcal{P}_\alpha$. The fact that $\mathcal{P}_\alpha$ is an $\alpha$-packing implies that the sets $B_P$ are all disjoint, and hence

$$1 \geq \sum_{P \in \mathcal{P}_\alpha} \Pr_{\hat{H} \leftarrow M(P_0^n)}\left[\hat{H} \in B_P\right] \geq |\mathcal{P}_\alpha| \cdot e^{-\varepsilon n} \cdot 9/10.$$

Rearranging gives us the stated lower bound on $n$. $\qquad\square$

**Lemma 5.2.** *For a set of distributions $\mathcal{H}$, let $p_\alpha$ and $c_\alpha$ be the size of the largest $\alpha$-packing and smallest $\alpha$-cover of $\mathcal{H}$, respectively. Then*

$$p_{2\alpha} \leq c_\alpha \leq p_\alpha.$$

*Proof.* We first prove the inequality on the left. Let $\mathcal{C}_\alpha$ be a cover of $\mathcal{H}$ of size $c_\alpha$. If $c_\alpha = \infty$, we are done. Otherwise, let $S$ be any set of points of size at least $c_\alpha + 1$. By the pigeonhole principle, there exists $P \in \mathcal{C}_\alpha$ and two distributions $Q, Q' \in S$ such that $d_{\mathrm{TV}}(P, Q) \leq \alpha$ and $d_{\mathrm{TV}}(P, Q') \leq \alpha$. Hence $d_{\mathrm{TV}}(Q, Q') \leq 2\alpha$ by the triangle inequality, so $S$ cannot be $(2\alpha)$-packing of $\mathcal{H}$. This suffices to show that $p_{2\alpha} \leq c_\alpha$.

Next, we prove the inequality on the right. Let $\mathcal{P}_\alpha$ be a maximal $\alpha$-packing with size $|\mathcal{P}_\alpha| = p_\alpha$. If $p_\alpha = \infty$, we are done. Otherwise, we claim that $\mathcal{P}_\alpha$ is also an $\alpha$-cover of $\mathcal{H}$, and hence $c_\alpha \leq |\mathcal{P}_\alpha| = p_\alpha$. To see this, suppose for the sake of contradiction that there were a distribution $P \in \mathcal{H}$ with $d_{\mathrm{TV}}(P, \mathcal{P}_\alpha) > \alpha$. Then we could add $P$ to $\mathcal{P}_\alpha$ to produce a strictly larger packing, contradicting the maximality of $\mathcal{P}_\alpha$. $\qquad\square$

**Theorem 5.3.** *Let $\mathcal{H}$ be a set of distributions, and let $n_\alpha^*$ denote the minimum number of samples such that there exists an $\varepsilon$-differentially private algorithm which takes as input samples $X_1, \ldots, X_{n_\alpha^*} \sim P$ for some $P \in \mathcal{H}$ and outputs a distribution $\hat{H}$ such that $d_{\mathrm{TV}}(P, \hat{H}) \leq \alpha$ with probability $\geq 9/10$. Then there exists a cover of $\mathcal{H}$ such that the instantiation of the algorithm underlying Theorem 1.1 with this cover outputs a $\hat{H}$ such that $d_{\mathrm{TV}}(P, \hat{H}) \leq 7\alpha$ with probability $\geq 9/10$ for any $n = \Omega(n_\alpha^* \cdot (\varepsilon/\alpha^2 + 1/\alpha))$.*

*Proof.* Let $p_\alpha$ denote the size of the largest $\alpha$-packing of $\mathcal{H}$. By Lemma 5.1, we have $n_\alpha^* = \Omega(\log p_\alpha/\varepsilon)$. On the other hand, by Lemma 5.2, we know that there exists an $\alpha$-cover $\mathcal{C}_\alpha$ of $\mathcal{H}$ with $|\mathcal{C}_\alpha| \leq p_\alpha$. Hence $\log |\mathcal{C}_\alpha| \leq O(\varepsilon \cdot n_\alpha^*)$ and the asserted sample complexity guarantee follows from Corollary 1.2. □

# 6 Applications of Hypothesis Selection

In this section, we give a number of applications of Theorem 1.1, primarily to obtain sample complexity bounds for learning a number of distribution classes of interest. Recall Corollary 1.2, which is an immediate corollary of Theorem 1.1. This indicates that we can privately semi-agnostically learn a class of distributions with a number of samples proportional to the logarithm of its covering number.

**Corollary 1.2.** *Suppose there exists an $\alpha$-cover $\mathcal{C}_\alpha$ of a set of distributions $\mathcal{H}$, and that we are given a set of samples $X_1, \ldots, X_n \sim P$, where $d_{\mathrm{TV}}(P, \mathcal{H}) \leq \alpha$. For any constant $\zeta > 0$, there exists an $\varepsilon$-differentially private algorithm (with respect to the input $\{X_1, \ldots, X_n\}$) which outputs a distribution $H^* \in \mathcal{C}_\alpha$ such that $d_{\mathrm{TV}}(P, H^*) \leq (6 + 2\zeta)\alpha$ with probability $\geq 9/10$, as long as*

$$n = \Omega\left(\frac{\log |\mathcal{C}_\alpha|}{\alpha^2} + \frac{\log |\mathcal{C}_\alpha|}{\alpha\varepsilon}\right).$$

Note that the factor of $(6 + 2\zeta)\alpha$ in the corollary statement (versus $(3 + \zeta)\alpha$ in the statement of Theorem 1.1) is due to the fact the algorithm is semi-agnostic, and the closest element in the cover is $2\alpha$-close to $P$, rather than just $\alpha$-close.

We instantiate this result to give the sample complexity results for semi-agnostically learning product distributions (Section 6.1), Gaussian distributions (Section 6.2), sums of some independent random variable classes (Section 6.3), piecewise polynomials (Section 6.4), and mixtures (Section 6.5). Furthermore, we mention an application to private PAC learning (Section 6.6), when the distribution of unlabeled examples is known to come from some hypothesis class.

## 6.1 Product Distributions

As a first application, we first give an $\varepsilon$-differentially private algorithm for learning product distributions over discrete alphabets.

**Definition 6.1.** *A $(k, d)$-product distribution is a distribution over $[k]^d$, such that its marginal distributions are independent (i.e., the distribution is the product of its marginals).*

We start by constructing a cover for product distributions.

**Lemma 6.2.** *There exists an $\alpha$-cover of the set of $(k, d)$-product distributions of size*

$$O\left(\frac{kd}{\alpha}\right)^{d(k-1)}.$$

*Proof.* Consider some fixed product distribution $P$, with marginal distributions $(P_1, \ldots, P_d)$. We will construct a cover that contains a distribution $Q$ (with marginals $(Q_1, \ldots, Q_d)$) that is $\alpha$-close in total variation distance.

First, by triangle inequality, we have that $d_{\mathrm{TV}}(P, Q) \leq \sum_{i=1}^d d_{\mathrm{TV}}(P_i, Q_i)$, so it suffices to approximate each marginal distribution to accuracy $\alpha/d$. Stated another way, we must generate an $(\alpha/d)$-cover of distributions over $[k]$, and we can then take its $d$-wise Cartesian product. Raising the size of this underlying cover to the power $d$ gives us the size of the overall cover.

To $(\alpha/d)$-cover a distribution over $[k]$, we will additively grid the probability of each symbol at granularity $\Theta\left(\frac{\alpha}{kd}\right)$, choosing the probability of the last symbol $k$ such that the sum is normalized. This will incur $\Theta\left(\frac{\alpha}{kd}\right)$ error per symbol (besides for symbol $k$), and summing over the $k-1$ symbols accumulates error $\Theta\left(\frac{\alpha}{d}\right)$. It can also be argued that the error on symbol $k$ is $O\left(\frac{\alpha}{d}\right)$ – with an appropriate choice of granularity, this gives us an $(\alpha/d)$-cover for distributions over $[k]$. The size of this cover is $O\left(\frac{kd}{\alpha}\right)^{k-1}$, which allows us to conclude the lemma statement. $\square$

With this cover in hand, applying Corollary 1.2 allows us to conclude the following sample complexity upper bound.

**Corollary 6.3.** *Suppose we are given a set of samples $X_1, \ldots, X_n \sim P$, where $P$ is $\alpha$-close to a $(k,d)$-product distribution. Then for any constant $\zeta > 0$, there exists an $\varepsilon$-differentially private algorithm which outputs a $(k,d)$-product distribution $H^*$ such that $d_{\mathrm{TV}}(P, H^*) \leq (6 + 2\zeta)\alpha$ with probability $\geq 9/10$, so long as*

$$n = \Omega\left(kd \log\left(\frac{kd}{\alpha}\right)\left(\frac{1}{\alpha^2} + \frac{1}{\alpha\varepsilon}\right)\right).$$

This gives the first $\tilde{O}(d)$ sample algorithm for learning a binary product distribution in total variation distance under pure differential privacy, improving upon the work of Kamath, Li, Singhal, and Ullman [KLSU19] by strengthening the privacy guarantee at a minimal cost in the sample complexity. The natural way to adapt their result from concentrated to pure differential privacy would require $\Omega(d^{3/2})$ samples.

**Remark 6.4.** *Properly learning a product distribution over $\{0,1\}^d$ to total variation distance $\leq \frac{1}{2}$ implies learning its mean $\mu \in [0,1]^d$ up to $\ell_1$ error $\leq 2\sqrt{d}$; see Lemma 6.5 below.*

*Thus Corollary 6.3 implies a $\varepsilon$-differentially private algorithm which takes $n = \tilde{O}(d/\varepsilon)$ samples from a product distribution $P$ on $\{0,1\}^d$ and, with high probability, outputs an estimate $\hat{\mu}$ of its mean $\mu$ with $\|\hat{\mu} - \mu\|_1 \leq 2\sqrt{d}$.*

*In contrast, for non-product distributions over the hypercube, estimating the mean to the same accuracy under $\varepsilon$-differential privacy requires $n = \Omega(d^{3/2}/\varepsilon)$ samples [HT10, SU15]. Thus we have a polynomial separation between estimating product and non-product distributions under pure differential privacy.*

**Lemma 6.5.** *If $P$ and $Q$ are product distributions on $\mathbb{R}^d$ with $d_{\mathrm{TV}}(P, Q) \leq \frac{1}{2}$ and per-coordinate variance at most $\sigma^2$, then*
$$\|\mathbf{E}_{X \leftarrow P}[X] - \mathbf{E}_{X \leftarrow Q}[X]\|_1 \leq 4\sqrt{d\sigma^2}.$$

*Proof.* Let $\mu = \mathbf{E}_{X \leftarrow P}[X] \in \mathbb{R}^d$ and $\mu' = \mathbf{E}_{X \leftarrow Q}[X] \in \mathbb{R}^d$. Let $\tau = \|\mu - \mu'\|_1$. Let $\nu = \mathrm{sign}(\mu - \mu') \in$

$\{-1, +1\}^d$ so that $\langle \nu, \mu - \mu' \rangle = \tau$. We have

$$
\begin{aligned}
\frac{1}{2} \geq d_{\mathrm{TV}}(P, Q) &\geq \Pr_{X \leftarrow P}[\langle \nu, X \rangle \geq t] - \Pr_{X \leftarrow Q}[\langle \nu, X \rangle \geq t] \\
&= \Pr_{X \leftarrow P}[\langle \nu, X - \mu \rangle \geq t - \langle \nu, \mu \rangle] - \Pr_{X \leftarrow Q}[\langle \nu, X - \mu' \rangle \geq t - \langle \nu, \mu \rangle + \langle \nu, \mu - \mu' \rangle] \\
(\text{set } t = \langle \nu, \mu \rangle - \frac{\tau}{2}) \quad &= \Pr_{X \leftarrow P}\left[\langle \nu, X - \mu \rangle \geq -\frac{\tau}{2}\right] - \Pr_{X \leftarrow Q}\left[\langle \nu, X - \mu' \rangle \geq +\frac{\tau}{2}\right] \\
&= 1 - \Pr_{X \leftarrow P}\left[\langle \nu, X - \mu \rangle < -\frac{\tau}{2}\right] - \Pr_{X \leftarrow Q}\left[\langle \nu, X - \mu' \rangle \geq +\frac{\tau}{2}\right] \\
(\text{Chebyshev's inequality}) \quad &\geq 1 - \frac{\mathbf{E}_{X \leftarrow P}[\langle \nu, X - \mu \rangle^2]}{(\tau/2)^2} - \frac{\mathbf{E}_{X \leftarrow Q}[\langle \nu, X - \mu' \rangle^2]}{(\tau/2)^2} \\
&= 1 - \frac{4}{\tau^2} \sum_{i=1}^{d} \mathbf{E}_{X \leftarrow P}[(X_i - \mu_i)^2] + \mathbf{E}_{X \leftarrow Q}[(X_i - \mu'_i)^2] \\
&\geq 1 - \frac{8d\sigma^2}{\tau^2}.
\end{aligned}
$$

Rearranging yields $\tau \leq 4\sqrt{d\sigma^2}$, as required. $\qquad \square$

## 6.2  Gaussian Distributions

We next give private algorithms for learning Gaussian distributions.

**Definition 6.6.** *A* Gaussian distribution $\mathcal{N}(\mu, \Sigma)$ *in* $\mathbb{R}^d$ *is a distribution with PDF*

$$
p(x) = \frac{\exp\left(-\frac{1}{2}(x - \mu)^T \Sigma^{-1}(x - \mu)\right)}{\sqrt{(2\pi)^d |\Sigma|}}.
$$

We describe covers for Gaussian distributions with known and unknown covariance.

**Lemma 6.7.** *There exists an $\alpha$-cover of the set of Gaussian distributions $\mathcal{N}(\mu, I)$ in $d$ dimensions with $\|\mu\|_2 \leq R$ of size*

$$
O\left(\frac{dR}{\alpha}\right)^d.
$$

*Proof.* It is well-known that estimating a Gaussian distribution with unknown mean in total variation distance corresponds to estimating $\mu$ in $\ell_2$-distance (see, e.g., [DKK$^+$16]). By the triangle inequality, in order to $\alpha$-cover the space, it suffices to $(\alpha/d)$-cover each standard basis direction. Since we know the mean in each direction is bounded by $R$, a simple additive grid in each direction with granularity $\Theta\left(\frac{\alpha}{d}\right)$ will suffice, resulting in a cover for each direction of size $O\left(\frac{dR}{\alpha}\right)$. Taking the Cartesian product over $d$ dimensions gives the desired result. $\qquad \square$

**Lemma 6.8.** *There exists an $\alpha$-cover of the set of Gaussian distributions $\mathcal{N}(\mu, \Sigma)$ in $d$-dimensions with $\|\mu\|_2 \leq R$ and $I \preceq \Sigma \preceq \kappa I$ of size*

$$
O\left(\frac{dR}{\alpha}\right)^d \cdot O\left(\frac{d\kappa}{\alpha}\right)^{d(d+1)/2}.
$$

*Proof.* The former term is obtained similarly to the expression in Lemma 6.7. Since $I \preceq \Sigma$, we can still bound the total variation contribution by the $\ell_2$-distance between the mean vectors. We thus turn our attention to the latter term. To construct our cover, we must argue about the total

variation distance between $\mathcal{N}(0, \Sigma)$ and $\mathcal{N}(0, \hat{\Sigma})$. If $|\Sigma(i, j) - \hat{\Sigma}(i, j)| \leq \gamma$, and $I \preceq \Sigma$, Proposition 32 of [VV10] implies:

$$d_{\mathrm{TV}}(\mathcal{N}(0, \Sigma), \mathcal{N}(0, \hat{\Sigma})) \leq O(d\gamma).$$

We will thus perform a gridding, in order to approximate each entry of $\Sigma$ to an additive $O(\gamma) = O(\alpha/d)$. However, in order to ensure that the resulting matrix is PSD, we grid over entries of $\hat{\Sigma}$'s Cholesky decomposition, rather than grid for $\hat{\Sigma}$ itself. Since the largest element of $\Sigma$ is bounded by $\kappa$, the larest element of its Cholesky decomposition must be bounded by $\sqrt{\kappa}$. An additive grid over the range $[0, \sqrt{\kappa}]$ with granularity $O(\gamma/\sqrt{\kappa})$ suffices to get $\hat{\Sigma}$ which bounds the entrywise distance as $O(\gamma)$. This requires $O(d\kappa/\alpha)$ candidates per entry, and we take the Cartesian product over all $d(d+1)/2$ entries of the Cholesky decomposition, giving the desired result. $\qquad\square$

In addition, we can obtain bounds of the VC dimension of the Scheffé sets of Gaussian distributions.

**Lemma 6.9.** *The set of Gaussian distributions with fixed variance – i.e., all $\mathcal{N}(\mu, I)$ with $\mu \in \mathbb{R}^d$ – has VC dimension $d + 1$. Furthermore, the set of Gaussians with unknown variance – i.e., all $\mathcal{N}(\mu, \Sigma)$ with $\mu \in \mathbb{R}^d$ and $\Sigma \in \mathbb{R}^{d \times d}$ positive definite – has VC dimension $O(d^2)$.*

*Proof.* For Gaussians with fixed variance, the Scheffé sets correspond to linear threshold functions, which have VC dimension $d + 1$. For Gaussians with unknown variance, the Scheffé sets correspond to quadratic threshold functions, which have VC dimension $\binom{d+2}{2} = O(d^2)$ [Ant95]. $\qquad\square$

Combining the covers of Lemmas 6.7 and 6.8 and the VC bound of Lemma 6.9 with Theorem 4.1 implies the following corollaries for Gaussian estimation.

**Corollary 6.10.** *Suppose we are given a set of samples $X_1, \ldots, X_n \sim P$, where $P$ is $\alpha$-close to a Gaussian distribution $\mathcal{N}(\mu, I)$ in $d$-dimensions with $\|\mu\| \leq R$. Then for any constant $\zeta > 0$, there exists an $\varepsilon$-differentially private algorithm which outputs a Gaussian distribution $H^*$ such that $d_{\mathrm{TV}}(P, H^*) \leq (6 + 2\zeta)\alpha$ with probability $\geq 9/10$, so long as*

$$n = \Omega\left(\frac{d}{\alpha^2} + \frac{d}{\alpha\varepsilon}\log\left(\frac{dR}{\alpha}\right)\right).$$

**Corollary 6.11.** *Suppose we are given a set of samples $X_1, \ldots, X_n \sim P$, where $P$ is $\alpha$-close to a Gaussian distribution $\mathcal{N}(\mu, \Sigma)$ in $d$-dimensions with $\|\mu\| \leq R$ and $I \preceq \Sigma \preceq \kappa I$. Then for any constant $\zeta > 0$, there exists an $\varepsilon$-differentially private algorithm which outputs a Gaussian distribution $H^*$ such that $d_{\mathrm{TV}}(P, H^*) \leq (6 + 2\zeta)\alpha$ with probability $\geq 9/10$, so long as*

$$n = \Omega\left(\frac{d^2}{\alpha^2} + \frac{1}{\alpha\varepsilon}\left(d\log\left(\frac{dR}{\alpha}\right) + d^2\log\left(\frac{d\kappa}{\alpha}\right)\right)\right).$$

Similar to the product distribution case, these are the first $\tilde{O}(d)$ and $\tilde{O}(d^2)$ sample algorithms for learning Gaussians total variation distance under pure differential privacy, improving upon the concentrated differential privacy results of Kamath, Li, Singhal, and Ullman [KLSU19].

### 6.2.1 Gaussians with Unbounded Mean

Extending Corollary 6.10, we consider multivariate Gaussian hypotheses with known covariance and unknown mean, *without* assuming bound on the mean (the parameter $R$ in the discussion above). To handle the unbounded mean we must relax to approximate differential privacy.

In place of Lemma 6.7, we construct a locally small cover:

**Lemma 6.12.** *For any $d \in \mathbb{N}$ and $\alpha \in (0, 1/30]$, there exists an $\alpha$-cover $\mathcal{C}_\alpha$ of the set of Gaussian distributions $\mathcal{N}(\mu, I)$ in $d$ dimensions satisfying*

$$\forall \mu \in \mathbb{R}^d \quad |\{H \in \mathcal{C}_\alpha : d_{\mathrm{TV}}(H, \mathcal{N}(\mu, I)) \leq 7\alpha\}| \leq 2^{15d}.$$

*Proof.* For $\mu, \mu' \in \mathbb{R}^d$, we have

$$d_{\mathrm{TV}}(\mathcal{N}(\mu, I), \mathcal{N}(\mu', I)) = 2\Pr\left[\mathcal{N}(0,1) \in \left[0, \frac{1}{2}\|\mu - \mu'\|_2\right]\right]$$

$$= \sqrt{\frac{2}{\pi}} \int_0^{\frac{1}{2}\|\mu - \mu'\|_2} e^{-x^2/2} \mathrm{d}x$$

$$\leq \frac{\|\mu - \mu'\|_2}{\sqrt{2\pi}}.$$

Furthermore, for any $c > 0$,

$$d_{\mathrm{TV}}(\mathcal{N}(\mu, I), \mathcal{N}(\mu', I)) \geq \begin{cases} \frac{\|\mu - \mu'\|_2}{\sqrt{2\pi}} \cdot e^{-c^2/2} & \text{if } \frac{1}{2}\|\mu - \mu'\|_2 \leq c \\ \frac{c \cdot e^{-c^2/2}}{\sqrt{2\pi}} & \text{if } \frac{1}{2}\|\mu - \mu'\|_2 \geq c \end{cases}.$$

Let

$$\mathcal{C}_\alpha = \left\{\mathcal{N}\left(m \cdot \frac{\alpha\sqrt{8\pi}}{\sqrt{d}}, I\right) : m \in \mathbb{Z}^d\right\}.$$

Fix $\mu \in \mathbb{R}^d$. Let $\mu^* = \mu \frac{\sqrt{d}}{\alpha\sqrt{8\pi}} \in \mathbb{R}^d$ and let $m \in \mathbb{Z}^d$ be $\mu^*$ rounded to the nearest integer coordinate-wise, so that $\|m - \mu^*\|_\infty \leq \frac{1}{2}$. Then

$$d_{\mathrm{TV}}\left(\mathcal{N}(\mu, I), \mathcal{N}\left(m \cdot \frac{\alpha\sqrt{8\pi}}{\sqrt{d}}, I\right)\right) = d_{\mathrm{TV}}\left(\mathcal{N}\left(\mu^* \cdot \frac{\alpha\sqrt{8\pi}}{\sqrt{d}}, I\right), \mathcal{N}\left(m \cdot \frac{\alpha\sqrt{8\pi}}{\sqrt{d}}, I\right)\right)$$

$$\leq \frac{1}{\sqrt{2\pi}} \frac{\alpha\sqrt{8\pi}}{\sqrt{d}} \|\mu^* - m\|_2$$

$$\leq \alpha,$$

since $\|\mu^* - m\|_2 \leq \sqrt{d}\|\mu^* - m\|_\infty \leq \frac{\sqrt{d}}{2}$. This proves that $\mathcal{C}_\alpha$ is a $\alpha$-cover of $\{\mathcal{N}(\mu, I) : \mu \in \mathbb{R}^d\}$.

It remains to show that the cover is "locally small". Let $m' \in \mathbb{Z}^d$. Then

$$d_{\mathrm{TV}}\left(\mathcal{N}(\mu, I), \mathcal{N}\left(m' \cdot \frac{\alpha\sqrt{8\pi}}{\sqrt{d}}, I\right)\right) = d_{\mathrm{TV}}\left(\mathcal{N}\left(\mu^* \cdot \frac{\alpha\sqrt{8\pi}}{\sqrt{d}}, I\right), \mathcal{N}\left(m' \cdot \frac{\alpha\sqrt{8\pi}}{\sqrt{d}}, I\right)\right)$$

$$\geq \frac{c \cdot e^{-c^2/2}}{\sqrt{2\pi}} \quad \text{if } \frac{1}{2}\|\mu^* - m'\|_2 \frac{\alpha\sqrt{8\pi}}{\sqrt{d}} \geq c$$

$$> 7\alpha \quad \text{if } \|\mu^* - m'\|_2 \geq 30\frac{\sqrt{d}}{\sqrt{2\pi}},$$

where the final inequality follows by setting $c = 30\alpha \leq 1$. Thus

$$
\begin{aligned}
\left|\{H \in \mathcal{C}_\alpha : d_{\mathrm{TV}}(H, \mathcal{N}(\mu, I)) \leq 7\alpha\}\right| &\leq \left|\left\{m' \in \mathbb{Z}^d : \|\mu^* - m'\|_2 < 30\frac{\sqrt{d}}{\sqrt{2\pi}}\right\}\right| \\
&\leq \left|\left\{m' \in \mathbb{Z}^d : \|m - m'\|_2 < 30\frac{\sqrt{d}}{\sqrt{2\pi}} + \|\mu^* - m'\|_2\right\}\right| \\
&\leq \left|\left\{m' \in \mathbb{Z}^d : \|m - m'\|_2 < 13\sqrt{d}\right\}\right| \\
&\leq \left|\left\{w \in \mathbb{Z}^d : \|w\|_1 < 13d\right\}\right|.
\end{aligned}
$$

Now we note that any $w \in \mathbb{Z}^d$ with $\|w\|_1 \leq r$ can be written as $w = x - y$ where $x, y \in \mathbb{Z}^d$ with $\sum_{i=1}^d x_i + y_i = r$ and, for all $i \in [d]$, we have $x_i \geq 0$ and $y_i \geq 0$. Instead of counting these $w$ vectors, we can count such $(x, y)$ vector pairs. We can interpret a pair of $x, y$ vectors as a way of putting $r$ balls into $2d$ bins or $r$ "stars" and $2d - 1$ "bars". We can thus count

$$
\left|\left\{w \in \mathbb{Z}^d : \|w\|_1 < 13d\right\}\right| \leq \left|\left\{x, y \in \mathbb{Z}^d : \|x\|_1 + \|y\|_2 = 13d - 1, x \geq 0, y \geq 0\right\}\right| \leq \binom{15d - 2}{2d - 1} \leq 2^{15d}.
$$

$\square$

Applying Theorem 4.1 with the cover of Lemma 6.12 and the VC bound from Lemma 6.9 now yields an algorithm.

**Corollary 6.13.** *Suppose we are given a set of samples $X_1, \ldots, X_n \sim P$, where $P$ is a spherical Gaussian distribution $\mathcal{N}(\mu, I)$ in $d$-dimensions. Then there exists a $(\varepsilon, \delta)$-differentially private algorithm which outputs a spherical Gaussian distribution $H^*$ such that $d_{\mathrm{TV}}(P, H^*) \leq 7\alpha$ with probability $\geq 1 - 2^{-d}$, so long as*

$$
n = \Omega\left(\frac{d}{\alpha^2} + \frac{d + \log(1/\delta)}{\alpha\varepsilon}\right).
$$

Karwa and Vadhan [KV18] give an algorithm for estimating a univariate Gaussian with un-bounded mean. One can consider applying their algorithm independently to the $d$ coordinates (which is done in [KLSU19]), giving a sample complexity bound of $\tilde{O}\left(\frac{d}{\alpha^2} + \frac{d}{\alpha\varepsilon} + \frac{\sqrt{d}\log^{3/2}(1/\delta)}{\varepsilon}\right)$, which our bound dominates except for very small values of $\alpha$.

### 6.2.2 Univariate Gaussians with Unbounded Mean and Variance

Our methods also allow us to derive learning algorithms for univariate Gaussians with unknown mean and variance.

**Lemma 6.14.** *For all $\alpha$ less then some absolute constant, there exists an $\alpha$-cover $\mathcal{C}_\alpha$ of the set of univariate Gaussian distributions satisfying*

$$
\forall \mu, \sigma \in \mathbb{R} \quad \left|\left\{H \in \mathcal{C}_\alpha : d_{\mathrm{TV}}(H, \mathcal{N}(\mu, \sigma^2)) \leq 7\alpha\right\}\right| \leq O(1).
$$

*Proof.* For all $\mu, \tilde{\mu} \in \mathbb{R}$ and all $\sigma, \tilde{\sigma} > 0$, we have [DMR18, Thm 1.3]

$$
\frac{1}{200}\min\left\{1, \max\left\{\frac{|\tilde{\sigma}^2 - \sigma^2|}{\tilde{\sigma}^2}, \frac{40|\tilde{\mu} - \mu|}{\tilde{\sigma}}\right\}\right\} \leq d_{\mathrm{TV}}(\mathcal{N}(\mu, \sigma^2), \mathcal{N}(\tilde{\mu}, \tilde{\sigma}^2)) \leq \frac{3|\tilde{\sigma}^2 - \sigma^2|}{2\tilde{\sigma}^2} + \frac{|\tilde{\mu} - \mu|}{2\tilde{\sigma}}.
$$

Let $\beta = \alpha$ and $\gamma = \log(1 + \alpha/2)$. Define the set of distributions

$$\mathcal{C}_\alpha = \left\{\mathcal{N}\left(\beta e^{\gamma n} m, e^{2\gamma n}\right) : n, m \in \mathbb{Z}\right\}.$$

We first show that $\mathcal{C}_\alpha$ is an $\alpha$-cover: Let $\mu \in \mathbb{R}$ and $\sigma > 0$. Let $n = \left[\frac{\log \sigma}{\gamma}\right]$ and $m = \left[\frac{\mu}{\beta e^{\gamma n}}\right]$, where $[x]$ denotes the nearest integer to $x$, satisfying $|x - [x]| \leq \frac{1}{2}$. Let $\tilde{\sigma} = e^{\gamma n}$ and $\tilde{\mu} = \beta e^{\gamma n} m$ so that $e^{-\gamma} \leq \frac{\tilde{\sigma}^2}{\sigma^2} \leq e^{\gamma}$ and $|\mu - \tilde{\mu}| \leq \frac{1}{2}\beta e^{\gamma n} = \frac{1}{2}\beta\tilde{\sigma}$. Thus $\mathcal{N}(\tilde{\mu}, \tilde{\sigma}^2) \in \mathcal{C}_\alpha$ and $d_{\mathrm{TV}}(\mathcal{N}(\mu, \sigma^2), \mathcal{N}(\tilde{\mu}, \tilde{\sigma}^2)) \leq \frac{3}{2}(e^\gamma - 1) + \frac{\beta}{4} \leq \alpha$, as required.

It only remains to show that the cover size is locally small. Let $\mu \in \mathbb{R}$ and $\sigma > 0$.

$$\left|\left\{H \in \mathcal{C}_\alpha : d_{\mathrm{TV}}(H, \mathcal{N}(\mu, \sigma^2)) \leq 7\alpha\right\}\right| = \left|\left\{n, m \in \mathbb{Z} : d_{\mathrm{TV}}(\mathcal{N}\left(\beta e^{\gamma n} m, e^{2\gamma n}\right), \mathcal{N}(\mu, \sigma^2)) \leq 7\alpha\right\}\right|$$

$$\leq \left|\left\{n, m \in \mathbb{Z} : \max\left\{\frac{|e^{2\gamma n} - \sigma^2|}{e^{2\gamma n}}, \frac{40|\beta e^{\gamma n} m - \mu|}{e^{\gamma n}}\right\} \leq 1400\alpha\right\}\right|$$

$$= \left|\left\{n, m \in \mathbb{Z} : \begin{array}{c}\frac{-\log(1+1400\alpha)}{2\gamma} \leq n - \frac{\log \sigma}{\gamma} \leq \frac{-\log(1-1400\alpha)}{2\gamma} \\ -35\frac{\alpha}{\beta} \leq m - \frac{\mu}{\beta e^{\gamma n}} \leq 35\frac{\alpha}{\beta}\end{array}\right\}\right|$$

$$\leq \left(\frac{-\log(1 - 1400\alpha)}{2\gamma} - \frac{-\log(1 + 1400\alpha)}{2\gamma} + 1\right) \cdot (35 - (-35) + 1)$$

$$= \frac{1}{2\log(1 + \alpha/2)} \log\left(\frac{1 + 1400\alpha}{1 - 1400\alpha}\right) \cdot 71 + 71$$

$$= O(1).$$

$\square$

Combining Lemma 6.14 with Lemma 6.9 and Theorem 4.1 yields the following.

**Corollary 6.15.** *Suppose we are given a set of samples $X_1, \ldots, X_n \sim P$, where $P$ is a univariate Gaussian distribution $\mathcal{N}(\mu, \sigma^2)$. Then there exists a $(\varepsilon, \delta)$-differentially private algorithm which outputs a univariate Gaussian distribution $H^*$ such that $d_{\mathrm{TV}}(P, H^*) \leq 7\alpha$ with probability $\geq 9/10$, so long as*

$$n = \Omega\left(\frac{1}{\alpha^2} + \frac{\log(1/\delta)}{\alpha\varepsilon}\right).$$

This sample complexity is comparable to to that of Karwa and Vadhan [KV18], who give an $(\varepsilon, \delta)$-DP algorithm with sample complexity $\tilde{O}\left(\frac{1}{\alpha^2} + \frac{1}{\alpha\varepsilon} + \frac{\log(1/\delta)}{\varepsilon}\right)$.

## 6.3 Sums of Independent Random Variables

In this section, we apply our results to distribution classes which are defined as the sum of independent (but not necessarily identical) distributions. These are all generalizations of the classical Binomial distribution, and they have enjoyed a great deal of study into the construction of sparse covers. To the best of our knowledge, we are the first to provide private learning algorithms for these classes.

We start with the Poisson Binomial distribution.

**Definition 6.16.** *A $k$-Poisson Binomial Distribution (k-PBD) is the sum of $k$ independent Bernoulli random variables.*

We next consider sums of independent integer random variables, which generalize PBDs (which correspond to the case $d = 2$).

**Definition 6.17.** *A $(k,d)$-Sum of Independent Integer Random Variables $((k,d)$-SIIRV) is the sum of $k$ independent random variables over $\{0,\dots,d-1\}$.*

Finally, we consider Poisson Multinomial distributions, which again generalize PBDs (which, again, correspond to the case $d = 2$).

**Definition 6.18.** *A $(k,d)$-Poisson Multinomial Distribution $((k,d)$-PMD) is the sum of $k$ independent $d$-dimensional categorical random variables, i.e., distributions over $\{e_1,\dots,e_d\}$, where $e_i$ is the $i$th basis vector.*

We start with a covering result for SIIRVs (including the special case of PBDs), which appears in [DKS16b]. Previous covers for PBDs and SIIRVs appear in [DP09, DP15b, DDO$^+$13].

**Lemma 6.19** ([DKS16b])**.** *There exists an $\alpha$-cover of the set of $(k,d)$-SIIRVs of size*

$$k \cdot 2^{O(d\log^2(1/\alpha)+d\log^2 d)}.$$

Using this cover, we can apply Corollary 1.2 to attain the following learning result for PBDs and SIIRVs.

**Corollary 6.20.** *Suppose we are given a set of samples $X_1,\dots,X_n \sim P$, where $P$ is $\alpha$-close to a $(k,d)$-SIIRV. Then for any constant $\zeta > 0$, there exists an $\varepsilon$-differentially private algorithm which outputs a $(k,d)$-SIIRV $H^*$ such that $d_{\mathrm{TV}}(P,H^*) \le (6+2\zeta)\alpha$ with probability $\ge 9/10$, so long as*

$$n = \Omega\left(\left(\log k + d\log^2(1/\alpha) + d\log^2 d\right)\left(\frac{1}{\alpha^2} + \frac{1}{\alpha\varepsilon}\right)\right).$$

Next, we move on to PMDs. The following cover does not appear verbatim in any single location, but is a combination of results from a few different sources. The proofs for the best bounds on first term appears in [DDKT16], the second in [DKT15], and the third in [DKS16a]. Larger covers previously appeared in [DP08, DP15a].

**Lemma 6.21** ([DKT15, DDKT16, DKS16a])**.** *For any $d > 2$, there exists an $\alpha$-cover of the set of $(k,d)$-PMDs of size*

$$k^{O(d)} \cdot \min\left\{2^{\mathrm{poly}(d/\alpha)}, (1/\alpha)^{O(d\log(d/\alpha)/\log\log(d/\alpha))^{d-1}}\right\}.$$

This implies the following learning result for PMDs.

**Corollary 6.22.** *Suppose we are given a set of samples $X_1,\dots,X_n \sim P$, where $P$ is $\alpha$-close to a $(k,d)$-PMD, for any $d > 2$. Then there exists an $\varepsilon$-differentially private algorithm which outputs a $(k,d)$-PMD $H^*$ such that $d_{\mathrm{TV}}(P,H^*) \le (6+2\zeta)\alpha$ with probability $\ge 9/10$, so long as*

$$n = \tilde{\Omega}\left(\left(d\log k + \min\left\{\mathrm{poly}\left(\frac{d}{\alpha}\right), O\left(\frac{d\log(d/\alpha)}{\log\log(d/\alpha)}\right)^{d-1} \cdot \log(1/\alpha)\right\}\right)\left(\frac{1}{\alpha^2} + \frac{1}{\alpha\varepsilon}\right)\right).$$

## 6.4 Piecewise Polynomials

In this section, we apply our results to semi-agnostically learn piecewise polynomials. This class of distributions is very expressive, allowing us to approximate a wide range of natural distribution classes.

**Definition 6.23.** *A $(t, d, k)$-piecewise polynomial distribution is a distribution $P$ over $[k]$, such that there exists a partition of $[k]$ into $t$ disjoint intervals $I_1, \ldots, I_t$ such that on each interval $I_j \subseteq [k]$, the probability mass function of $P$ takes the form $p_j(x) = \sum_{i=0}^{d} c_i^{(j)} x^i$ for some coefficients $c_i^{(j)}$, for all $x \in I_j$.*

We construct a cover for piecewise polynomials.

**Lemma 6.24.** *There exists a universal constant $c > 0$ such that there is an $\alpha$-cover of the set of $(t, d, k)$-piecewise polynomials of size*

$$\binom{k}{t-1} \cdot \left( \frac{tk \cdot e^{cd^{1/2}}}{\alpha} \right)^{(d+1)t}.$$

*Proof.* We specify an element of the cover by

1. Selecting one of $\binom{k}{t-1}$ partitions of $[k]$ into $t$ intervals $I_1, \ldots, I_t$, and

2. For each interval $I_j$, selecting an element of an $(\alpha/t)$-cover $\mathcal{C}_j$ of the set of degree-$d$ polynomials over $I_j$ which are uniformly bounded by 1.

The total size of the cover is $\binom{k}{t-1} \prod_{j=1}^{t} |\mathcal{C}_j|$. The theorem follows from Proposition 6.25 below, which constructs an $(\alpha/t)$-cover $\mathcal{C}_j$ of size at most $\left( \frac{tk \cdot e^{cd^{1/2}}}{\alpha} \right)^{d+1}$ for every interval $I_j$. $\qquad \square$

**Proposition 6.25.** *There exist constants $b, c > 0$ for which the following holds. Let $I \subseteq [k]$ be an interval and let $\mathcal{P}$ be the set of polynomials $p : I \to \mathbb{R}$ of degree $d$ such that $|p(x)| \leq 1$ for all $x \in I$. There exists an $\alpha$-cover of $\mathcal{P}$ of size*

$$\min \left\{ \left( \frac{2k}{\alpha} \right)^{|I|}, \left( \frac{ckd^2 \cdot e^{bd^2/|I|}}{\alpha} \right)^{d+1} \right\}.$$

The proof of Proposition 6.25 relies on two major results in approximation theory, which we now state.

**Lemma 6.26** (Duffin and Schaeffer [DS41]). *Let $p : [-1, 1] \to \mathbb{R}$ be a polynomial such that $|p(x)| \leq 1$ for all $x$ of the form $x = \cos(j\pi/d)$ for $j = 0, 1, \ldots, d$. Then $|p'(x)| \leq d^2$ for all $x \in [-1, 1]$.*

**Lemma 6.27** (Coppersmith and Rivlin [CR92]). *There exist constants $a, b > 0$ for which the following holds. Let $p : \mathbb{R} \to \mathbb{R}$ be a polynomial of degree $d$, and suppose that $|p(t)| \leq 1$ for all $t = 0, 1, \ldots, m$. Then $|p(t)| \leq a \exp(bd^2/m)$ for all $t \in [0, m]$.*

*Proof of Proposition 6.25.* We consider two cases, corresponding to the two terms in the minimum. First, consider the function $f : I \to \mathbb{R}$ where $f(t)$ is obtained by rounding $p(t)$ to the nearest multiple of $\alpha/k$. Then $f$ satisfies $\sum_{t \in I} |f(t) - p(t)| \leq \alpha$. There are at most $(2k/\alpha)^{|I|}$ functions $f$ which can be constructed this way, giving the first term in the maximum.

For the second term, we construct a cover for $\mathcal{P}$ by approximately interpolating through $d + 1$ carefully chosen points in the *continuous* interval corresponding to $I$. By applying an affine shift, we may assume that $I = \{0, 1, \ldots, m\}$ for some integer $m \leq k - 1$. Let $p \in \mathcal{P}$ and for $x \in [0, m]$ let $\hat{p}(x)$ be the value of $p(x)$ rounded to the nearest integer multiple of $\alpha/(2kd^2)$. Let $q : [0, m] \to \mathbb{R}$

be the unique degree-$d$ polynomial obtained by interpolating through the points $(x_j, \hat{p}(x_j))$ where $x_j = m(1 + \cos(j\pi/d))/2$ for $j = 0, 1, \ldots, d$.

We first argue that the polynomial $q$ so defined satisfies $\sum_{t=0}^{m} |p(t) - q(t)| \leq \alpha$. Let $r(x) = p(x) - q(x)$ for $x \in [0, m]$. Then by construction, $|r(x_j)| \leq \alpha/(2kd^2)$ for all interpolation points $x_j$. By the Duffin-Schaeffer Inequality (Lemma 6.26), we therefore have $|r'(x)| \leq \frac{\alpha}{km}$ for all $x \in [0, m]$. By the Fundamental Theorem of Calculus, $r(t) = r(0) + \int_0^t r'(t)\,dt$ satisfies $|r(t)| \leq (t+1) \cdot \frac{\alpha}{km} \leq \alpha/k$, and hence $\sum_{t=0}^{m} |r(t)| \leq \alpha$.

We now argue that the set of polynomials $q$ that can be constructed in this fashion has size $(ckd^2 \exp(bd^2/m)/\alpha)^{d+1}$. By the Coppersmith-Rivlin Inequality (Lemma 6.27), there are constants $a, b > 0$ such that $|p(x)| \leq a\exp(bd^2/m)$ for all $x \in [0, m]$. Therefore, for each $p \in \mathcal{P}$ and each interpolation point $x_j$, there are at most $4a \cdot kd^2 \exp(bd^2/m)/\alpha$ possible values that $\hat{p}(x_j)$ can take. Hence, the polynomial $q$ can take one of at most $(4a \cdot kd^2 \exp(bd^2/m)/\alpha)^{d+1}$ possible values, as we wanted to show. $\qquad\square$

**Lemma 6.28.** *The VC dimension of $(t, d, k)$-piecewise polynomial distributions is at most $2t(d+1)$.*

*Proof.* Consider two piecewise polynomial distributions. The difference between their probability mass functions is a piecewise polynomial of degree $\leq d$. The number of intervals needed to represent this piecewise function is $\leq 2t$. It follows that this difference can change sign at most $2td + 2t - 1$ times – each polynomial can change sign at most $d$ times and the sign can change at the interval boundaries. Thus such a function cannot label $2td + 2t + 1$ points with alternating signs, which implies the VC bound. $\qquad\square$

As a corollary, we obtain the following learning algorithm.

**Corollary 6.29.** *Suppose we are given a set of samples $X_1, \ldots, X_n \sim P$, where $P$ is $\alpha$-close to a $(t, d, k)$-piecewise polynomial. Then there exists an $\varepsilon$-differentially private algorithm which outputs a $(t, d, k)$-piecewise polynomial $H^*$ such that $d_{\mathrm{TV}}(P, H^*) \leq (6 + 2\zeta)\alpha$ with probability $\geq 9/10$, so long as*

$$n = \Omega\left(\frac{(d+1)t}{\alpha^2} + \frac{(d+1)t}{\alpha\varepsilon} \cdot \left(\sqrt{d+1}\log k + \log\left(\frac{t}{\alpha}\right)\right)\right).$$

We compare with the work of Diakonikolas, Hardt, and Schmidt [DHS15]. They present an efficient algorithm for $(t, 1, k)$-piecewise polynomials, with sample complexity $\tilde{O}\left(\frac{t}{\alpha^2} + \frac{t\log k}{\alpha\varepsilon}\right)$, which our algorithm matches[6]. They also claim their results extend to $(t, d, k)$-piecewise polynomials, though no theorem statement is provided. While we have not investigated the details of this extension, we believe the resulting sample complexity should be qualitatively similar to ours, plausibly with the factor of $\frac{t(d+1)^{3/2}\log k}{\alpha\varepsilon}$ replaced by $\frac{t(d+1)\log k}{\alpha\varepsilon}$.

## 6.5 Mixtures

In this section, we show that our results immediately extend to learning mixtures of classes of distributions.

**Definition 6.30.** *Let $\mathcal{H}$ be some set of distributions. A $k$-mixture of $\mathcal{H}$ is a distribution with density $\sum_{i=1}^{k} w_i P_i$, where each $P_i \in \mathcal{H}$.*

Our results follow roughly due to the fact that a cover for $k$-mixtures of a class can be written as the Cartesian product of $k$ covers for the class. More precisely, we state the following result which bounds the size of the cover of the set of $k$-mixtures.

**Lemma 6.31.** *Consider the class of $k$-mixtures of $\mathcal{H}$, where $\mathcal{H}$ is some set of distributions. There exists a $2\alpha$-cover of this class of size $|\mathcal{C}_\alpha|^k \left(\frac{k}{2\alpha} + 1\right)^{k-1}$, where $\mathcal{C}_\alpha$ is an $\alpha$-cover of $\mathcal{H}$.*

*Proof.* Each element in the cover of the class of mixtures will be obtained by taking $k$ distributions from $\mathcal{C}_\alpha$, in combination with $k$ mixing weights, which are selected from the set $\left\{0, \frac{2\alpha}{k}, \frac{4\alpha}{k}, \ldots, 1\right\}$, such that the sum of the mixing weights is 1. The size of this cover is $|\mathcal{C}_\alpha|^k \cdot \left(\frac{k}{2\alpha} + 1\right)^{k-1}$. We reason about the accuracy of the cover as follows. Fix some mixture of $k$ distributions as $\sum_{i=1}^{k} w_i^{(1)} P_i^{(1)}$, and we will reason about the closest element in our cover, $\sum_{i=1}^{k} w_i^{(2)} P_i^{(2)}$. By triangle inequality, we have that

$$d_{\mathrm{TV}} \left( \sum_{i=1}^{k} w_i^{(1)} P_i^{(1)}, \sum_{i=1}^{k} w_i^{(2)} P_i^{(2)} \right) \leq \sum_{i=1}^{k} \frac{1}{2} \left| w_i^{(1)} - w_i^{(2)} \right| + w_i^{(1)} d_{\mathrm{TV}} \left( P_i^{(1)}, P_i^{(2)} \right).$$

Since $\mathcal{C}_\alpha$ is an $\alpha$-cover and $\sum_{i=1}^{k} w_i^{(1)} = 1$, the total variation distance incurred by the second term will be at most $\alpha$. As for the mixing weights, note that for the first $k - 1$ weights, the nearest weight is at distance at most $\frac{\alpha}{k}$, contributing a total of less than $\frac{\alpha}{2}$. The last mixing weight can be rewritten in terms of the sum of the errors of the other mixing weights, similarly contributing another total of less than $\frac{\alpha}{2}$. This results in the total error being at most $2\alpha$, as desired. $\square$

With this in hand, the following corollary is almost immediate from Corollary 1.2. The factor of $(9 + 3\zeta)\alpha$ (as opposed to $(6 + 2\zeta)\alpha$) is because the closest distribution in the cover of mixture distributions is $3\alpha$-close to be $P$ (rather than $2\alpha$).

**Corollary 6.32.** *Let $X_1, \ldots, X_n \sim P$, where $P$ is $\alpha$-close to a $k$-mixture of distributions from some set $\mathcal{H}$. Let $\mathcal{C}_\alpha$ be an $\alpha$-cover of the set $\mathcal{H}$, and $\zeta > 0$ be a constant. There exists an $\varepsilon$-differentially private algorithm which outputs a distribution which is $(9+3\zeta)\alpha$-close to $P$ with probability $\geq 9/10$, as long as*

$$n = \Omega \left( (k \log |\mathcal{C}_\alpha| + k \log(k/\alpha)) \left( \frac{1}{\alpha^2} + \frac{1}{\alpha\varepsilon} \right) \right).$$

For example, instantiating this for mixtures of Gaussians (and disregarding terms which depend on $R$ and $\kappa$), we get an algorithm with sample complexity $\tilde{O}\left( \frac{kd^2}{\alpha^2} + \frac{kd^2}{\alpha\varepsilon} \right)$.

## 6.6 Supervised Learning

We describe an application of our results to the task of binary classification, as modeled by differentially private PAC learning [KLN+11]. Let $\mathcal{F} = \{f : X \to \{0, 1\}\}$ be a publicly known *concept class* of Boolean functions over a domain $X$. Let $P$ be an unknown probability distribution over $X$, and let $f$ be an unknown function from $\mathcal{F}$. Given a sequence $\{(x_i, f(x_i))\}_{i=1}^{n}$ of i.i.d. samples from $P$ together with their labels under $f$, the goal of a PAC learner $L$ is to identify a hypothesis $h : X \to \{0, 1\}$ such that $\Pr_{x \sim P}[h(x) \neq f(x)] \leq \alpha$ for some error parameter $\alpha > 0$. We say that $L$ is $(\alpha, \beta)$-*accurate* if for every $f \in \mathcal{F}$ and every distribution $P$, it is able to identify such a hypothesis $h$ with probability at least $1 - \beta$ over the choice of the sample and any internal randomness of $L$.

One of the core results of statistical learning theory is that the sample complexity of *non-private* PAC learning is characterized, up to constant factors, by the VC dimension of the concept class $\mathcal{F}$.

When one additionally requires the learner $L$ to be differentially private with respect to its input sample, such a characterization is unknown. However, it is known that the sample complexity of private learning can be arbitrarily higher than that of non-private learning. For example, when $\mathcal{F} = \{f_t : t \in X\}$ is the class of threshold functions defined by $f_t(x) = 1 \iff x \leq t$ over a totally ordered domain $X$, the sample complexity of PAC learning under the most permissive notion of $(\varepsilon, \delta)$-differential privacy is $\Omega(\log^* |X|)$ [BNSV15, ALMM19]. Meanwhile, the VC dimension of this class, and hence the sample complexity of non-private learning, is a constant independent of $|X|$.

While this separation shows that there can be a sample cost of privacy for PAC learning, this cost can be completely eliminated if the distribution $P$ on examples is known. This was observed by Beimel, Nissim, and Stemmer [BNS16], who showed that if a good approximation to $P$ is known, e.g., from public unlabeled examples or from differentially private processing of unlabeled examples, then the number of labeled examples needed for private PAC learning is only $O(VC(\mathcal{F}))$.

**Theorem 6.33.** *Let $\varepsilon > 0$, $\mathcal{F} = \{f : X \to \{0,1\}\}$, and $P$ be a publicly known distribution over $X$. For $n = O\left(\frac{1}{\alpha^2 \varepsilon}(VC(\mathcal{F}) \log(1/\alpha) + \log(1/\beta))\right)$, there exists an $\varepsilon$-differentially private algorithm $L : (X \times \{0,1\})^n \to \mathcal{F}$ such that for every $f \in \mathcal{F}$, with probability at least $1 - \beta$ over the choice of $x_1, \ldots, x_n \leftarrow P$, we have that $L((x_1, f(x_1)), \ldots, (x_n, f(x_n)))$ produces $h \in \mathcal{F}$ such that $\Pr_{x \sim P}[f(x) \neq h(x)] \leq \alpha$.*

Our results suggest a natural two-step algorithm for private PAC learning when the distribution $P$ itself is not known, but is known to (approximately) come from a set of distributions $\mathcal{H}$: The algorithm first uses private hypothesis selection to select $\hat{H}$ with $d_{\mathrm{TV}}(P, \hat{H}) \leq \alpha/2$, and then runs the algorithm of [BNS16] using $\hat{H}$ in place of $P$ with error parameter $\alpha/2$. Using the fact that $d_{\mathrm{TV}}(P, \hat{H}) \leq \alpha/2$ implies $|\Pr_{x \sim P}[f(x) \neq h(x)] - \Pr_{x \sim \hat{H}}[f(x) \neq h(x)]| \leq \alpha/2$, the following result holds by combining Theorem 6.33 with Corollary 1.2.

**Corollary 6.34.** *Let $\mathcal{H}$ be a set of distributions over $X$ with an $\alpha$-cover $\mathcal{C}_\alpha$. Let $P$ be a distribution over $X$ with $d_{\mathrm{TV}}(P, \mathcal{H}) \leq \alpha/(4(3+\zeta))$. Then for*

$$n = O\left(\frac{\log |\mathcal{C}_\alpha|}{\alpha^2} + \frac{\log |\mathcal{C}_\alpha|}{\alpha \varepsilon} + \frac{VC(\mathcal{F}) \log(1/\alpha)}{\alpha^2 \varepsilon}\right)$$

*there exists an $\varepsilon$-differentially private algorithm $L : (X \times \{0,1\})^n \to \mathcal{F}$ such that for every $f \in \mathcal{F}$, with probability at least $3/4$ over the choice of $x_1, \ldots, x_n \leftarrow P$, we have that $L((x_1, f(x_1)), \ldots, (x_n, f(x_n)))$ produces $h \in \mathcal{F}$ such that $\Pr_{x \sim P}[f(x) \neq h(x)] \leq \alpha$.*

Theorem 6.33 can, of course, also be combined with the more refined guarantees of Theorem 4.1. As an example application, combining Theorem 6.33 with Corollary 6.13 gives a $(\varepsilon, \delta)$-differentially private algorithm for learning one-dimension thresholds with respect to univariate Gaussian distributions on the reals. In contrast, this task is impossible without making distributional assumptions.

# 7    Conclusions

In this paper, we presented differentially private methods for hypothesis selection. The sample complexity can be bounded by the logarithm of the number of hypotheses. This allows us to provide bounds on the sample complexity of (semi-agnostically) learning a class which depend on the logarithm of the covering number, complementing known lower bounds which depend on the logarithm of the packing number. There are many interesting questions left open by our work, a few of which we outline below.

1. Our algorithms for learning classes of distributions all use cover-based arguments, and thus are not computationally efficient. For instance, we provide the first $\tilde{O}(d)$ sample complexity upper bound on $\varepsilon$-differentially privately learning a product distribution and Gaussian with known covariance. One interesting question is whether there is an efficient algorithm which achieves this sample complexity.

2. The running time of our method is quadratic in the number of hypotheses – is it possible to reduce this to a near-linear time complexity?

3. Our main theorem obtains an approximation factor which is arbitrarily close to 3, which is optimal for this problem, even without privacy. This factor can be reduced to 2 if one is OK with outputting a mixture of hypotheses from the set [BKM19]. Is this achievable with privacy constraints?

# Acknowledgments

The authors would like to thank Shay Moran for bringing to their attention the application to PAC learning mentioned in Section 6.6, Jonathan Ullman for asking questions which motivated Remark 6.4, and Clément Canonne for assistance in reducing the constant factor in the approximation guarantee.

## Footnotes

[1]Roughly, this is due to the fact that the Laplace and Gaussian mechanism are based on $\ell_1$ and $\ell_2$ sensitivity, respectively, and that there is a $\sqrt{d}$-factor relationship between these two norms, in the worst case.

[2]Given two probability distributions $P, Q$ over $\Omega$, $R_{\alpha}(P \| Q) = \frac{1}{\alpha - 1} \log \left( \sum_{x \in \Omega} P(x)^{\alpha} Q(x)^{1 - \alpha} \right)$.

[3]For simplicity of our exposition, we will assume that we can evaluate the two quantities $p_1$ and $p_2$ exactly. In general, we can estimate these quantities to arbitrary accuracy, as long as, for each hypothesis $H$, we can evaluate the density of each point under $H$ and also draw samples from $H$.

[4]Here, for simplicity, we assume that each distribution $H$ is given by a density function $H(\cdot)$. More generally, we define the VC dimension of $\mathcal{H}$ to be the smallest $d$ such that there exists a function family $\mathcal{F} \subseteq \{0,1\}^{\mathcal{X}}$ of VC dimension $d$ with the property that, for all $H, H' \in \mathcal{H}$ we have $d_{\mathrm{TV}}(H, H') = \sup_{f \in \mathcal{F}} \mathbf{E}_{X \leftarrow H}[f(X)] - \mathbf{E}_{X \leftarrow H'}[f(X)]$, where the supremum is over $f$ measurable with respect to both $H$ and $H'$. We ignore this technicality throughout.

[5]It suffices for $G$ to be a drawn from a universal hash function family.

[6]As stated in [DHS15], their algorithm guarantees approximate differential privacy, but swapping in an appropriate pure DP subroutine gives this result.