[Reviews · NeurIPS 2019]

Reviewer 1



Overall, I think this is a well-written paper that provides a significant technical contribution on an important problem in differential privacy, and it would be a good addition to the NeurIPS program. Minor comments, mostly on presentation, are given below. --pg 2, Thm 1: does alpha need to be known in advance? If not (and alpha has to be chosen by the user), then what happens if no such H* exists? This is all made clear later in the technical sections, but should be clarified here. --pg 2, lines 57-61: For applications where m is so large that Omega(m) is inefficient, then this algorithms runtime of m^2 is even more infeasible. --pg 3, lines 90-92,103: Is d still the VC-based measure from a few paragraphs ago? Relatedly the earlier mention of Theorem 3 feels sufficiently separated that I thought the mention of it here was a typo. These paragraphs could be better linked together. --pg 5, lines 200-201: What does it mean that a distribution behaves like a function applied to a set? I can imagine when thinking of a distribution as a hypothesis as the average label over that set (although I'm not sure this is the correct interpretation since we're not talking about labels here), but then there is P(W_1) and P is the true distribution. When defining W_1, should \mathcal{X} be D? Confusion over these details also let to some confusion when interpreting the results. --pg 6, proof of Lemma 2: The claim that Gamma(H_1,H_2) has sensitivity 1 is not obvious and should be proven/argued. This confusion is related to the point above because Gamma is a piecewise-linear functions whose conditionals and values depend on the terms W_1, p_1, p_2, and tau that were unclear. --pg 6, line 263: "indepdently" --pg 7, line 267: concentrated differential privacy hasn't been mentioned yet. It should be either defined (formally or informally) in the body or include a pointer to a formal definition in the supplementary materials.

Reviewer 2



This paper is a solid contribution towards understand the sample complexity costs of achieving differential privacy during hypothesis testing. The paper explores this setting quite well, along with several motivating applications of a fairly abstract general mechanism. Originality: The scoring function that measure the number of points that must be changed to change a pairwise comparison of hypotheses appears to be a novel contribution, and its application into the exponential mechanism (or GAP-MAX) achieves novel utility bounds. The paper sufficiently cites related work in this commonly explored problem. Quality: The work appears quite sound with theoretical results, and the strengths and limitations of the analysis are explored. Clarity: The work is quite clearly written, carefully building up their mechanism and its analysis. Significance: The wide reach of their improved bounds to several applications is significant.

Reviewer 3



This work studies the problem of selecting a hypothesis from a class of hypotheses that is close in TV-distance from an unknown distribution P that samples are drawn from, with the added constraint of DP. The overall algorithm is the exponential mechanism, but some care needs to be taken in selecting the correct score function. The paper needs a lot more explanatory text to help the reader understand the intuition. Some things are defined with no explanation, such as \Gamma(H,H’,D). The paper dismisses in a footnote the issue that p_1 and p_2 can be evaluated exactly, because they can be estimated to arbitrary accuracy. How does an error in p_1 or p_2 propagate through the analysis? Overall I think the paper tackles an interesting problem and uses standard DP techniques to solve it. There are many different applications that the paper points out to show the impact of this result. The paper would benefit from a thorough proof reading with more explanations to improve the writing. This could all be fixed. Comments: - "indepdently" on page 2. - Should cite the various constraints studied previously in hypothesis selection on Page 1. - Lots of terms are used before they are defined, like DP, \alpha-cover. - The variable “d” is overloaded in Section 1.1 (refers to VC dim and I assume the dimension of the Gaussian distribution). - Preliminaries just state lots of definitions, with no explanatory text. - In Definition 3, how to define d_{TV}(P,\mathcal{H}) when d_{TV} is only defined over distributions. Also missing a \hat{H} in the d_{TV} bound. - Footnote on page 5 refers to H_j, but should this be H’? - Proof of Lemma 2, is H_j and H_k supposed to be H_1 and H_2? “probablity” on page 6 - In the Applications of Hypothesis Selection section, the notation O(x)^d is used. Is this supposed to be O(x^d)? ### I have read the author feedback and will keep my score unchanged.

[Author Response · NeurIPS 2019]

We would like to thank all the reviewers for their careful reading and constructive suggestions.

To address a common question of Reviewer #3 and #5: as currently stated in the paper, Theorem 1 applies only to the case when we have an upper bound on the target quality of distribution, and such a distribution exists. In the latest revision of the paper, we additionally describe an extension of this result which does not require this knowledge. More precisely, the algorithm we give is "semi-agnostic" – if OPT is the minimum total variation distance between the target distribution and any of the hypotheses, it outputs a hypothesis which has distance $O(\mathrm{OPT}) + \alpha$ from the target – at the cost of additive terms which are generally lower order (i.e., the sample complexity $n$ increases by an additive $O(\log \log(1/\alpha)(1/\alpha^2 + \log^2(1/\alpha)/(\alpha \epsilon)))$. We will describe this result in the final version, together with a complete proof in the supplementary material.

We will additionally be sure to address all of the detailed feedback from Reviewers 3 and 5, which we believe will significantly improve the presentation of the paper. Regarding three specific comments:

- Reviewer #3's question about probability distributions applied to a set: the notation $P(W_1)$ stands for the probability mass which distribution $P$ applies to the set $W_1$. This will be clarified in the next version.

- Reviewer #5's question about how error in $p_1$ and $p_2$ propagate in the analysis: It is sufficient to estimate these quantities up to an additive $O(\alpha)$, which can be done with $O(1/\alpha^2)$ samples to each of the candidate hypotheses. This will be rigorously argued in the next version.

- A near-linear time algorithm: While we made some attempts to derive a near-linear time algorithm, they were roadblocked by a number of known lower bounds for canonical problems (e.g., lower bounds for top-$k$ selection). As a result, we consider this to be an interesting avenue for further study – we weakly conjecture that there may be a deeper phenomenon at play, potentially including a trade-off between time and sample complexity.

[Meta-Review · NeurIPS 2019]

This work studies the differentially private hypothesis selection problem. Hypothesis selection problem is a workhorse of statistics/ML and these tasks are often performed on data that is privacy-sensitive, e.g. health data. This work studies the differentially private version of this question. A bit more precisely, given a set of hypotheses Q_1,...Q_k, and given samples from a distribution P, the goal is to find an (approximate) minimizer of the statistical distance between P and Q_i's. This is a well studied problem. The authors show sample complexity bounds to solve this problem privately, and show applications of this algorithm. Comments: The factor of 7 is worse than the classical non-private bound of 3. Can this be improved? Also, the factor of 3 has been improved to 2 for the improper case. Can your work be extended to improve the bound for the improper case.?